# miRNA-214-3p stimulates carcinogen-induced mammary epithelial cell apoptosis in mammary cancer-resistant species

Rebecca M. Harman [1], Sanjna P. Das[1], Matt Kanke[2], Praveen Sethupathy[2] & Gerlinde R. Van de Walle [1✉]

Mammary cancer incidence varies greatly across species and underlying mechanisms remain elusive. We previously showed that mammosphere-derived epithelial cells from species with low mammary cancer incidence, such as horses, respond to carcinogen 7, 12-Dimethyl-benz(a)anthracene-induced DNA damage by undergoing apoptosis, a postulated anti-cancer mechanism. Additionally, we found that miR-214-3p expression in mammosphere-derived epithelial cells is lower in mammary cancer-resistant as compared to mammary cancer-susceptible species. Here we show that increasing miR-214 expression and decreasing expression of its target gene nuclear factor kappa B subunit 1 in mammosphere-derived epithelial cells from horses abolishes 7,12-Dimethylbenz(a)anthracene-induced apoptosis. A direct interaction of miR-214-3p with another target gene, unc-5 netrin receptor A, is also demonstrated. We propose that relatively low levels of miR-214 in mammosphere-derived epithelial cells from mammals with low mammary cancer incidence, allow for constitutive gene nuclear factor kappa B subunit 1 expression and apoptosis in response to 7, 12-Dimethylbenz(a)anthracene. Better understanding of the mechanisms regulating cellular responses to carcinogens improves our overall understanding of mammary cancer resistance mechanisms.

[1] Baker Institute for Animal Health, College of Veterinary Medicine, Cornell University, Ithaca, NY 14853, USA. [2] Department of Biomedical Sciences, College of Veterinary Medicine, Cornell University, Ithaca, NY 14853, USA. ✉email: grv23@cornell.edu

Despite the mammary gland being conserved in structure and function, the incidence of mammary cancer varies greatly across species. While humans, dogs, cats, and ferrets are highly prone to mammary cancer, it is rarely documented in other mammals such as horses, pigs, cows, and cheetahs[1–3]. Growing evidence suggests that inherent variations at a cellular level may contribute to cancer resistance. Elephants, which are long-lived cancer-resistant wild mammals, have additional copies of the tumor suppressor gene p53, and their peripheral blood lymphocytes undergo increased p53-mediated apoptosis in response to irradiation-induced DNA damage when compared to human lymphocytes[4]. Fibroblasts from the naked mole rat, a long-lived rodent, secrete high molecular mass hyaluronan, which leads to early contact inhibition-mediated resistance to cancer[5,6]. Also, fibroblasts from blind mole rats undergo a necrotic cell death response triggered by pro-growth signals, another proposed strategy to avert cancer[7]. Our group previously demonstrated that mammary stem/progenitor cells (MaSCs) from mammals with varying mammary cancer incidence respond differently to carcinogen-induced DNA damage[8], supporting the theory that intrinsic differences in mammary cells may be responsible for variations in mammary cancer incidence across mammals. Specifically, we showed that the carcinogen 7, 12-Dimethylbenz(a)anthracene (DMBA) causes comparable degrees of DNA damage in MaSCs from mammals with high (e.g., dog) and low (e.g., horse) mammary cancer incidence. After DMBA-induced damage, canine MaSCs are repaired, allowing for the potential accumulation of oncogenic mutations, while equine MaSCs are eliminated by apoptosis[8]. These findings led us to propose a model in which mammals with low mammary cancer incidence reduce the risk of accumulating oncogenic mutations during DNA repair by eliminating mammary cells with DNA damage from the population. Intriguingly, this mechanism shows compelling similarities with the enhanced apoptotic response to DNA damage observed in elephant cells[4,9] and may function as an evolutionarily conserved protective mechanism in long-lived and cancer-resistant mammals.

MicroRNAs (miRNAs) are post-transcriptional regulators of gene expression that interact with target mRNAs in a sequence-specific manner, generally leading to mRNA degradation or translational repression. miRNAs play important roles in numerous developmental and growth processes, including apoptosis. They have been shown to regulate both the intrinsic (mitochondrial) and extrinsic apoptotic pathways, as well as apoptosis triggered by endoplasmic reticulum (ER) stress[10,11]. Specific miRNAs that regulate the intrinsic pathway have been associated with resistance to chemotherapeutics[12,13] or inhibition of tumor growth[14,15]. In the extrinsic pathway, some miRNAs have been shown to protect cells from programmed cell death[16–18], while others sensitize them to apoptosis[19], and several miRNAs that regulate the ER stress-induced pathway promote cell survival[20,21]. These mechanisms have been elucidated in different cells and tissues, with the majority in the context of cancer. Although the expression and function of miRNAs in breast cancer is an intense area of research, baseline miRNA expression patterns in healthy mammary glands have not been well-explored and the ways in which miRNAs support cellular processes in normal mammary epithelial cells are not well understood.

The goal of the present study was to determine if miRNAs play a role in equine MaSC apoptosis in response to DMBA-induced DNA damage. Since our previous study was published[8], we have renamed MaSCs as mammosphere-derived epithelial cells (MDECs). MaSCs and MDECs are isolated and cultured following the exact same protocol, the name change was implemented to more accurately describe the cell population based on the isolation method. Our findings in this study were that: (i) miR-214 is among the set of miRNAs that are significantly differentially expressed between canine and equine MDECs; (ii) increasing miR-214 levels or (iii) decreasing expression of the miR-214 target gene nuclear factor kappa B subunit 1 (NFKB1) prevents DMBA-induced apoptosis of equine MDECs; and (iv) miR-214-3p directly interacts with target gene UNC5A. Collectively, this work provides critical molecular insights into mechanisms in normal mammary epithelial cells that may confer resistance to mammary cancer.

## Results

**Canine and equine mammosphere-derived epithelial cells (MDECs) respond differently to 7, 12-Dimethylbenz(a) anthracene (DMBA) and have distinct miRNA expression patterns.** Corroborating our previous findings[8], treatment with the carcinogen DMBA at 5 μM did not affect the viability of canine MDEC cultures, while the viability of equine MDECs was significantly decreased (Fig. 1a). To determine if variations in miRNA expression profiles explain the differential responses of canine and equine MDECs to DMBA, we used our previously published microRNA (miRNA) sequencing data set, derived from MDEC cultures isolated from 6 different mammalian species[22] to evaluate the miRNA expression profiles between canine and equine MDECs (Fig. 1b). Fifty-three differentially expressed miRNAs were identified ($P_{adj} < 0.05$; Supplementary Data 1) of which we selected 4 miRNAs that are known to be involved in apoptosis[23–26]. Two miRNAs, miR-183-5p and miR-214-3p, were expressed more highly in canine compared to equine MDECs (Fig. 1c(i)), while the other two, miR-16-1-5p and miR-27a-3p were expressed more highly in equine compared to canine MDECs (Fig. 1c(ii)). We used quantitative reverse transcription-polymerase chain reaction (qRT-PCR) to validate the miRNA expression patterns detected by the miRNA sequencing and confirmed the expression patterns of miR-183 and miR-214 (Fig. 1d(i)), but not of miR-16-2 and miR-27a (Fig. 1d(ii)). Of the two miRNAs confirmed by qRT-PCR, we focused on miR-214, as this miRNA is more robustly expressed in MDECs than miR-183 (Fig. 1c) and is reported to mediate cellular processes in a normal murine mammary epithelial cells[24].

**Increasing miR-214 expression in equine MDECs prevents DMBA-induced apoptosis.** We used a miR-214 mimic; a small chemically modified double-stranded RNA that is commercially available and mimics endogenous miR-214, to increase miR-214 expression and activity in equine MDECs, to evaluate whether altered miR-214 expression affects apoptosis in response to DMBA. Transfection of equine MDECs with the miR-214 mimic resulted in a significantly increased expression of this miRNA when compared to non-transfected control cells and cells transfected with a negative mimic (Supplementary Fig. 1a(i)). A small, but significant, reduction in viability was observed in equine MDECs after transfection with both the miR-214 mimic and negative mimic when compared to control cells (Supplementary Fig. 1a(ii)), indicating that the transfection procedure affected cell viability, but only modestly. We next treated non-transfected equine MDECs (control), miR-214 mimic-transfected equine MDECs, and negative mimic-transfected equine MDECs, with either nothing (no treatment), DMBA, or the vehicle control dimethylsulfoxide (DMSO), and evaluated cell viability using MTT assays as well as apoptosis using flow cytometry to detect caspase activity and immunostaining to evaluate active caspase-3 expression. As expected, cell viability was significantly reduced after DMBA treatment in control equine MDECs and negative mimic-transfected equine MDECs when compared to their respective no-treatment and DMSO conditions (Fig. 2a(i)).

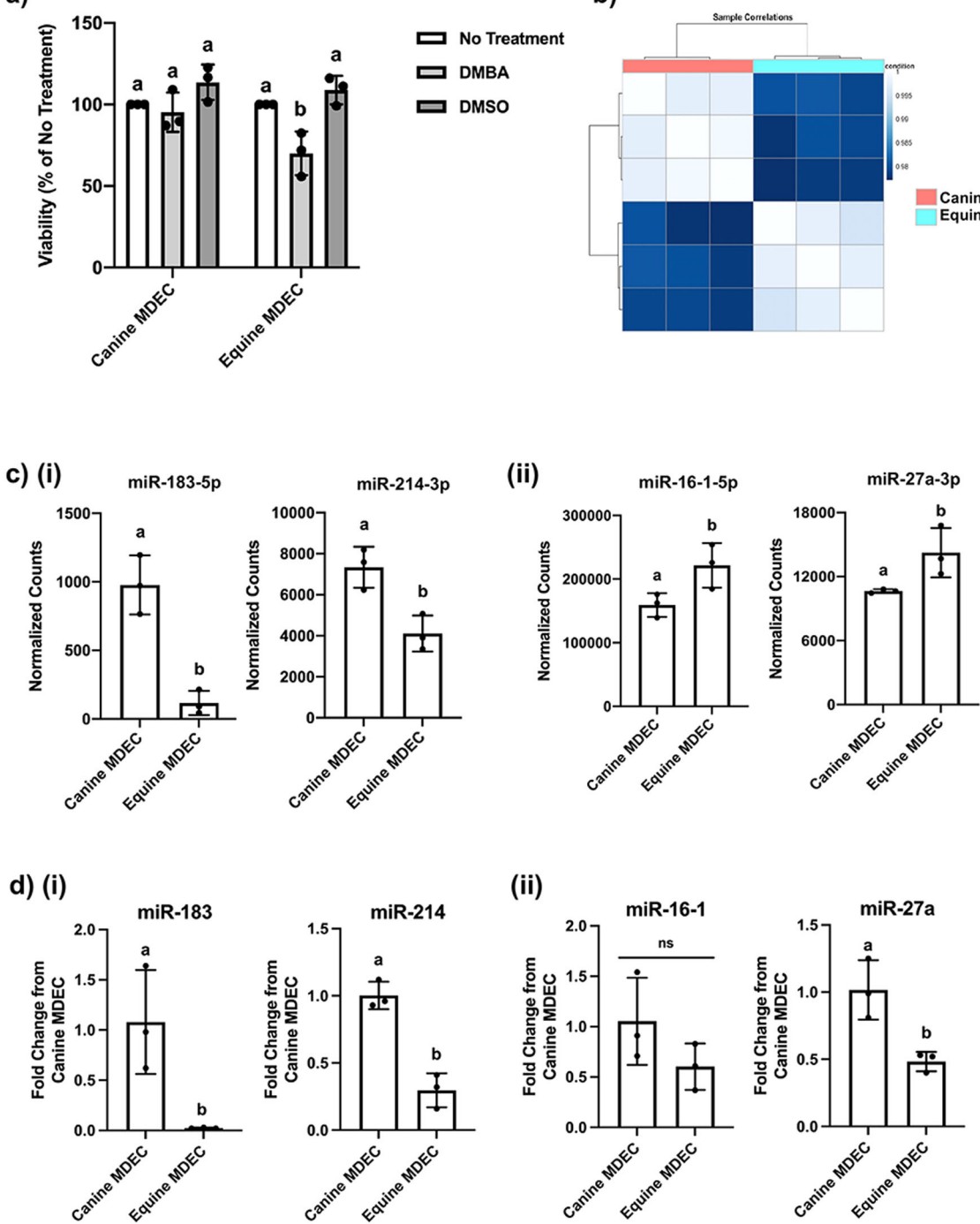

**Fig. 1 Canine and equine mammosphere-derived epithelial cells (MDECs) respond differently to 7, 12-Dimethylbenz(a)anthracene (DMBA) and have distinct microRNA (miRNA) expression patterns. a** Viability of canine and equine MDECs after treatment with 5 µM DMBA or the vehicle control, dimethylsulfoxide (DMSO). **b** Correlation matrix of miRNA expression in canine and equine MDECs ($n = 3$ individual cell cultures/species) showing that miRNA expression patterns in cells from the same species are more closely related than those from different species. **c** Normalized counts of miRNAs of interest that were detected at either (**i**) higher or (**ii**) lower expression levels in canine compared to equine MDEC in miRNA sequencing analysis. **d** Quantitative reverse transcription-polymerase chain reaction (qRT-PCR) analysis of the miRNAs of interest depicted in **c**. $n = 3$. Error bars show standard deviations. Different letters above the bars indicate statistically significant differences. $P < 0.05$.

Alternatively, the viability of miR-214 mimic-transfected equine MDECs did not significantly change after treatment with DMBA when compared to miR-214 mimic-transfected equine MDECs that received either no treatment or treatment with DMSO (Fig. 2a(i)). Rounded, dead cells were visible by eye in culture wells showing reduced viability (Fig. 2a(ii)). The level of cell

viability in DMBA-treated miR-214 mimic-transfected equine MDECs was corroborated by a lack of elevated caspase activity detected by flow cytometry (Fig. 2b), as well as the absence of active caspase-3 expression in these cells based on immunostaining (Fig. 2c). In contrast, caspase activity and active caspase-3-positive cells significantly increased in DMBA-treated control

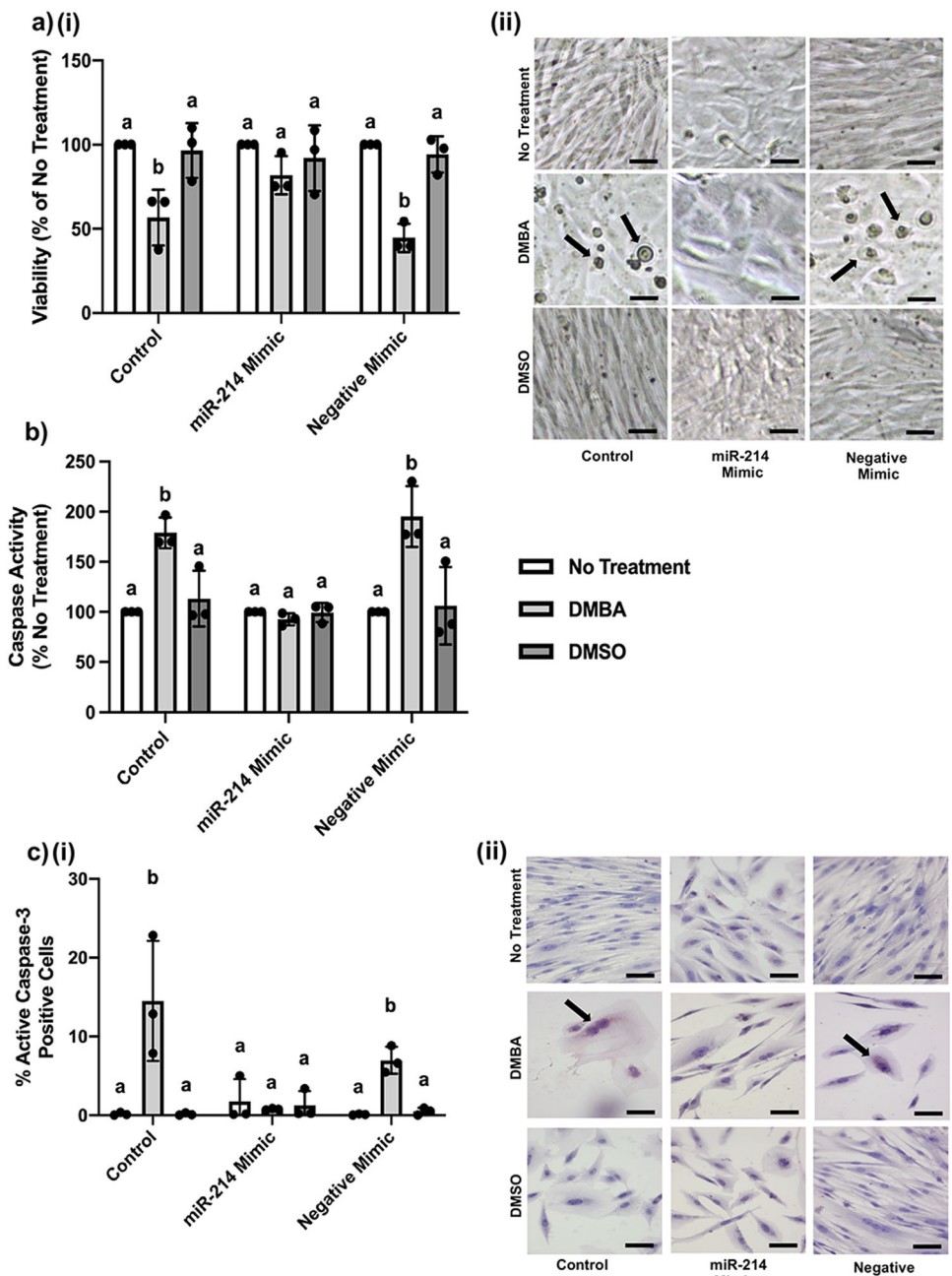

**Fig. 2 Increasing microRNA-214 (miR-214) in equine mammosphere-derived epithelial cells (MDECs) inhibits 7, 12-Dimethylbenz(a)anthracene (DMBA)-induced apoptosis. a** Viability of equine MDECs that were untransfected, transfected with a miR-214 mimic or transfected with a negative miRNA mimic, followed by no treatment, treatment with 5 μM DMBA, or treatment with the vehicle control dimethylsulfoxide (DMSO), as determined by MTT assays (**i**) and representative phase contrast images, with arrows pointing to dead cells (**ii**). **b** Apoptotic cells detected in equine MDECs that were untransfected, transfected with a miR-214 mimic or transfected with a negative miRNA mimic, followed by no treatment, treatment with 5 μM DMBA, or treatment with DMSO, as determined by flow cytometric detection of caspase activity. **c** Percentage active caspase-3 positive (apoptotic) cells per microscopic field in equine MDEC cultures that were untransfected, transfected with a miR-214 mimic, or transfected with a negative miRNA mimic, followed by no treatment, treatment with 5 μM DMBA, or treatment with DMSO, as determined by anti-active caspase-3 immunolabeling (**i**) and representative images, with arrows pointing to red active caspase-3-positive cells (**ii**). Scale bars = 20 μm. $n = 3$. Error bars show standard deviations. Different letters above the bars indicate statistically significant differences. $P < 0.05$.

and negative mimic-transfected equine, when compared to their respective no treatment and DMSO conditions (Fig. 2b, c). Collectively, these data show that increasing miR-214 expression in equine MDECs abolishes apoptosis in response to DMBA.

Although not the focus of this work, we questioned whether a similar, but opposite, effect could be observed by altering miR-214 expression in canine MDECs, which have an inherently

higher expression of this miRNA than equine MDECs (Fig. 1c(i), d(i)) and do not undergo apoptosis in response to DMBA (Fig. 1a). To determine this, we transfected canine MDECs with a miR-214 inhibitor; a small chemically modified single-stranded RNA that is commercially available and binds endogenous miR-214, inhibiting activity. We observed significantly reduced miR-214 expression when compared to non-transfected control and

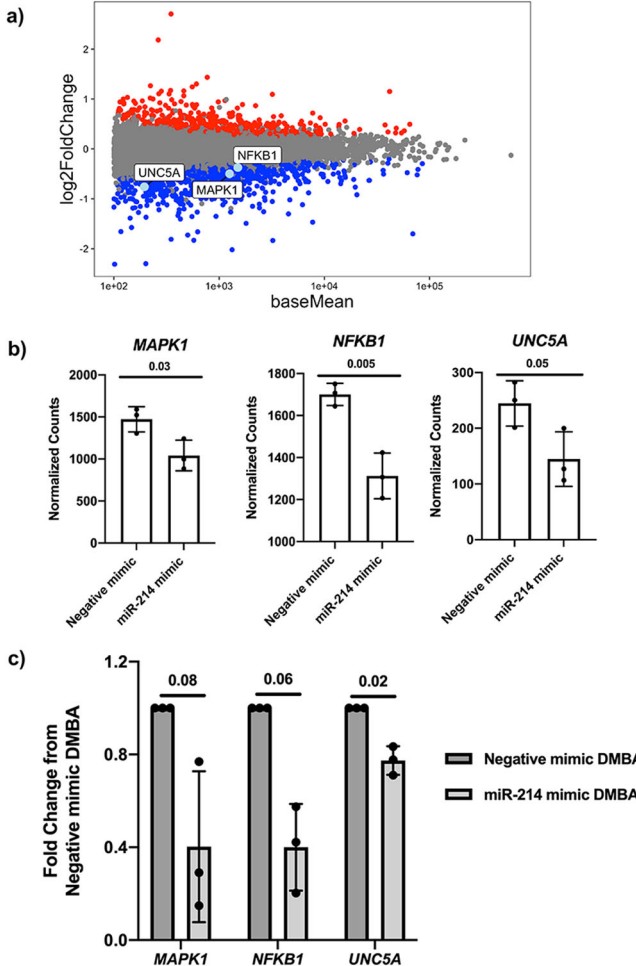

**Fig. 3 Increasing microRNA-214 (miR-214) in equine mammosphere-derived epithelial cells (MDECs) leads to the downregulation of predicted miR-214-3p target genes. a** MA plot showing differentially expressed genes (DEG) detected in equine MDECs transfected with a miR-214 mimic and treated with 7, 12-Dimethylbenz(a)anthracene (DMBA). **b** Normalized mRNA expression, and **c** quantitative reverse transcription-polymerase chain reaction (qRT-PCR) analysis of genes of interest that were detected at lower levels in equine MDECs transfected with a miR-214 mimic compared to equine MDECs transfected with a negative miRNA mimic and treated with DMBA. $n = 3$. Error bars show standard deviations. *P*-values are indicated on graphs.

negative inhibitor-transfected canine cells (Supplementary Fig. 1b(i)), and no statistically significant effect on cell viability after transfection with either the miR-214 or negative inhibitor (Supplementary Fig. 1b(ii)). Despite a significant reduction of miR-214 expression, miR-214-transfected canine MDECs did not undergo apoptosis in response to DMBA when compared to the no treatment and DMSO conditions, and cell viability resembled non-transfected and negative inhibitor-transfected canine MDECs (Supplementary Fig. 1c). Collectively, these data suggest that either miR-214 does not play a role in regulating DMBA-mediated apoptosis in canine MDECs or that the reduction in miR-214 expression was not sufficient to induce a DMBA-mediated apoptosis response in canine MDECs.

**Increasing miR-214 expression in equine MDECs results in downregulation of putative miR-214-3p target genes.** RNA deep sequencing (RNA-seq) was performed on equine MDECs transfected with the miR-214 mimic or the negative mimic,

followed by DMBA treatment (Supplementary Data 2), to determine which miR-214 target genes could be involved in DMBA-induced apoptosis. Differential expression analysis revealed approximately 807 down-regulated genes in miR-214 mimic-transfected DMBA-treated equine MDECs and of those, 100 genes are predicted miR-214-3p targets (Supplementary Table 1). Candidate target genes were prioritized based on (i) a functional role in apoptosis and (ii) an observed change in expression of genes downstream of the candidate target in the RNA-seq analysis. As the predominant effect of miRNAs is downregulation of target gene expression[27], a corresponding change in the expression of genes downstream of predicted miR-214 targets suggests that miR-214 may serve as a universal upstream regulator. Unc-5 netrin receptor A *(UNC5A)*, mitogen-activated protein kinase 1 *(MAPK1)*, and nuclear factor kappa B subunit 1 *(NFKB1)* met these criteria[27–31] (Fig. 3a, b and Supplementary Fig. 2) and were selected for functional follow-up studies. After confirming the downregulation of these three genes in miR-214 mimic-transfected DMBA-treated equine MDECs by qRT-PCR (Fig. 3c), we knocked down these predicted target genes in equine MDECs and assessed the effects of the knock-downs on DMBA-induced apoptosis.

**miR-214-3p directly interacts with UNC5A.** When considering the 3' untranslated regions (UTRs) of the three selected target genes, we observed that *UNC5A* has an exceptionally strong potential miR-214-3p binding site consisting of 13 consecutive base pairings (Fig. 4a). For comparison, the 3' UTRs of *MAPK1* and *NFKB1* have 9 and 6 consecutive pairings for miR-214-3p binding, respectively (Fig. 4a). *UNC5A* is not a previously validated target of miR-214-3p, which prompted us to evaluate whether miR-214 binds directly to the 3' UTR of *UNC5A*. To this end, we carried out dual-luciferase assays in HeLa cells, a commonly used cell line that lends itself well to transfection studies[32]. The suitability of Hela cells as a proxy for equine MDECs in the dual-luciferase assays was confirmed by qRT-PCR, showing endogenous miR-214 expression levels in both cell lines (Fig. 4b). Hela cells were transfected with an expression clone containing firefly luciferase (FLuc) and the 3'UTR of *UNC5A* under control of the SV40 promotor, as well as *Renilla* luciferase (RLuc) controlled by the CMV reporter (Fig. 4c). The ratio of FLuc/RLuc significantly decreased in the presence of the miR-214 mimic and significantly increased in the presence of the miR-214 inhibitor, as compared to cells transfected with just the expression clone (Fig. 4d(i)). This suggests that (i) the miR-214 mimic binds the 3' UTR of *UNC5A* in the plasmid, targeting FLuc for degradation and reduced expression, and (ii) the miR-214 inhibitor blocks endogenous cellular miR-214 expression, preventing it from binding the 3' UTR of *UNC5A* in the plasmid leading to increased expression. To support these findings, we repeated the dual-luciferase assay with an expression clone containing a 3-nucleotide deletion in the 3' UTR of *UNC5A* that corresponded with the seed region of the proposed miR-214-3p binding site. Transfecting cells with this expression clone no longer resulted in changes in FLuc/RLuc activity in the presence of the miR-214 mimic or inhibitor (Fig. 4d(ii)), indicating that the decreased FLuc signal we detected in the first assays (Fig. 4d(i)) was indeed the result of direct binding of miR-214 to the 3' UTR of *UNC5A*.

**Collective knockdown of UNC5A, MAPK1, and NFKB1 in equine MDECs inhibits DMBA-induced apoptosis, which was found to be primarily mediated by NFKB1.** To evaluate the potential functional role of the miR-214-3p target genes *UNC5A*, *MAPK*1, and/or *NFKB1*, in DMBA-induced apoptosis of equine MDECs, we used small-interfering RNAs (siRNAs) to knock down

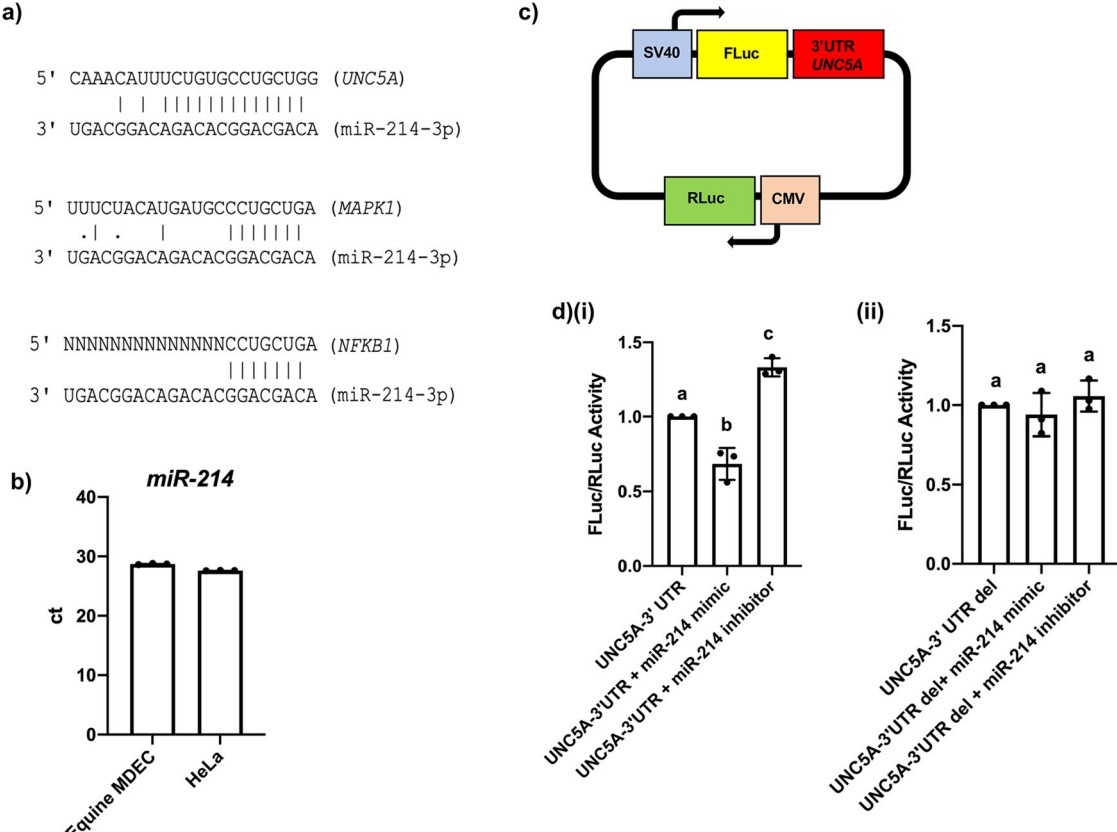

**Fig. 4 microRNA-214 (miR-214) binds to the 3′ untranslated region (UTR) of unc-5 netrin receptor A (*UNC5A*). a** Alignment of miR-214-3p with the 3′ UTRs of *UNC5A*, mitogen-activated protein kinase 1 (*MAPK1*), and nuclear factor kappa beta 1 (*NFKB1*). 'N' represents nucleotides in the open reading frame. **b** Quantitative reverse transcription-polymerase chain reaction (qRT-PCR) analysis of miR-214 expression in equine MDECs and HeLa cells. **c** Schematic of the *UNC5A* 3′ UTR expression clone used in luciferase assays. **d** Luciferase activity detected in HeLa cells transfected with the (**i**) wild type or (**ii**) mutant *UNC5A* 3′UTR expression plasmid either alone or together with the miR-214 mimic or the miR-214 inhibitor. $n = 3$. Error bars show standard deviations. Different letters above the bars indicate statistically significant differences. $P < 0.05$.

all three genes. We first confirmed that a cocktail of the 3 respective siRNAs was capable of significantly reducing the expression of all 3 genes and corresponding proteins when compared to their expression in non-transfected cells (Supplementary Fig. 3a). As expected, gene and protein expression was not significantly altered when using a negative siRNA at the same concentration as the siRNA cocktail (Supplementary Fig. 3a). We next treated non-transfected equine MDECs (control), triple siRNA-transfected equine MDECs, and negative siRNA-transfected equine MDECs, with either nothing (no treatment), 5 μM DMBA, or the vehicle control dimethylsulfoxide (DMSO). As expected, cell viability was significantly reduced after DMBA treatment in control equine MDECs and negative siRNA-transfected equine MDECs, when compared to their respective no-treatment and DMSO conditions (Fig. 5a(i)). However, no significant difference in cell viability of triple siRNA-transfected equine MDECs was observed after treatment with DMBA when compared to triple siRNA-transfected equine MDECs that received either no treatment or treatment with DMSO (Fig. 5a(i)). The observed reductions in cell viability corresponded with increases in apoptosis (Fig. 5a(ii)). To identify which of these genes was responsible for resistance to apoptosis in response to DMBA, we repeated these experiments with a single knock-down of either *UNC5A*, *MAPK1*, or *NFKB1*. RT-qPCR confirmed efficient knock-down of each gene and protein individually (Supplementary Fig. 3b, c, d). Representative images of Western blots probed with antibodies against *UNC5A*, *MAPK1*, or *NFKB* are shown in Supplementary Fig. 3e. Assessment of cell viability after DMBA treatment showed that knock-down of

*UNC5A* or *MAPK1* alone did not affect DMBA-induced apoptosis (Fig. 5b(i), c(i)), whereas knock-down of *NFKB1* alone inhibited the DMBA-induced reduction in cell viability observed in the control and negative siRNA-transfected equine MDECs (Fig. 5d(i)). Again, observed reductions in cell viability corresponded with increases in apoptosis (Fig. 5b(ii), c(ii), d(ii)). Brightfield images of cell cultures taken before viability assays were captured and are shown in Supplementary Fig. 4. Collectively, these data show that decreasing expression of the miR-214 target gene *NFKB1*, but not *UNC5A* or *MAPK1*, inhibits equine MDECs from undergoing DMBA-induced apoptosis.

**NFKB1 expression levels do not appear to modulate the resistance of canine MDECs to DMBA-induced cell death, and decreased NFKB1 expression in equine MDECs relies on the combination of increased miR-214 expression and DMBA treatment.** To provide insight into the role of *NFKB1* expression in MDECs, we performed the following set of experiments. Based on the observation that decreasing *NFKB1* in equine MDECs inhibits these cells from undergoing DMBA-induced apoptosis (Fig. 5d), we predicted that canine MDECs would inherently express lower levels of *NFKB1* compared to equine MDECs, as canine MDECs do not undergo DMBA-induced cell death (Fig. 1a)/apoptosis[8]. To evaluate this, degenerate PCR primers were designed that amplify both equine and canine *NFKB1* transcripts. Contrary to our prediction, canine MDECs constitutively expressed significantly more *NFKB1* compared to equine MDECs (Fig. 6a(i)), suggesting that a mechanism

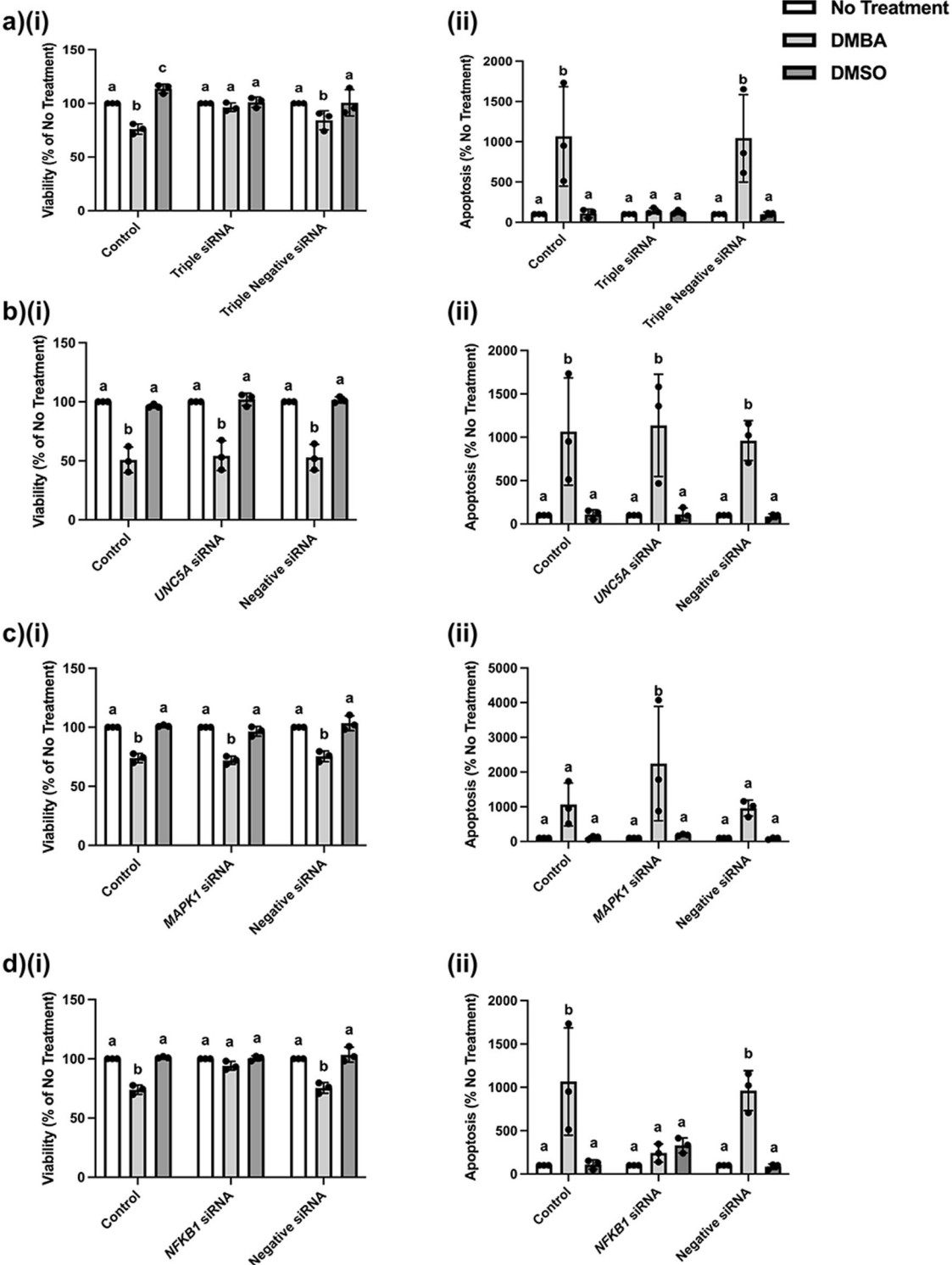

**Fig. 5 Knockdown of miR-214 target genes unc-5 netrin receptor A *(UNC5A)* or mitogen-activated protein kinase 1 *(MAPK)1* in equine mammosphere-derived epithelial cells (MDECs) does not inhibit cells from undergoing 7, 12-Dimethylbenz(a)anthracene (DMBA)-induced apoptosis, while knockdown of nuclear factor kappa beta 1 *(NFKB1)* does. a** Viability, as detected by MTT assay (**i**), and apoptosis as determined by flow cytometric detection of generic caspase activity (**ii**), of equine MDEC that were untransfected, transfected with a *MAPK1, NFKB1,* and *UNC5A* siRNA cocktail or transfected with a negative siRNA, followed by no treatment, treatment with 5 μM DMBA, or treatment with the vehicle control dimethylsulfoxide (DMSO). **b, c, d** Viability, as detected by MTT assay (**i**), and apoptosis as determined by flow cytometric detection of caspase activity (**ii**), of equine MDEC that were untransfected, transfected with either *UNC5A* siRNA (**b**), *MAPK1* siRNA (**c**), or *NFKB1* siRNA (**d**), or transfected with a negative siRNA, followed by no treatment, treatment with 5 μM DMBA, or treatment with the vehicle control DMSO. *n* = 3. Error bars show standard deviations. Different letters above the bars indicate statistically significant differences. *P* < 0.05. Note: the control and negative siRNA data presented in **c(i)** and **d(i)** are from the same experiment.

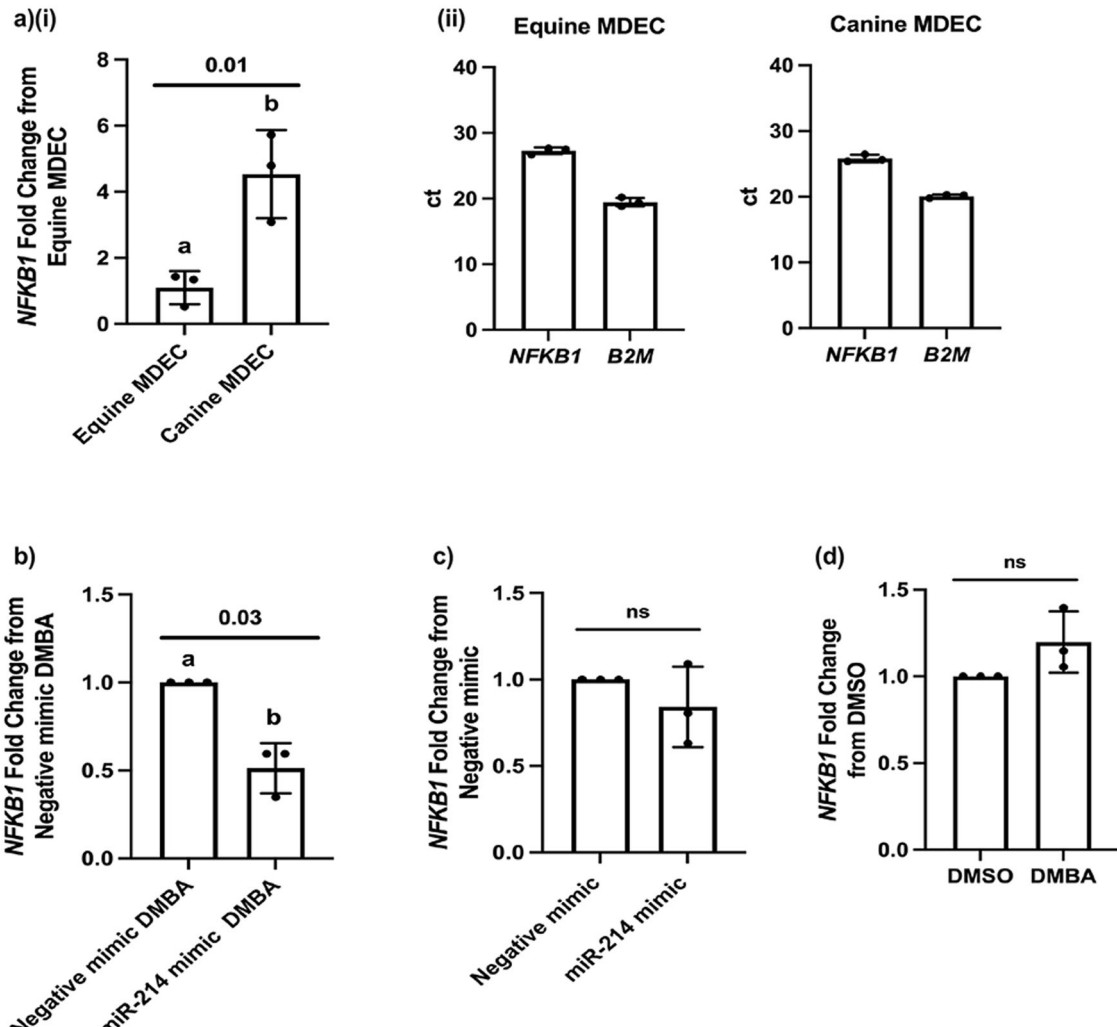

**Fig. 6 Nuclear factor kappa beta 1 (NFKB1) levels may not be involved in the resistance of canine mammosphere-derived epithelial cells (MDECs) to 7, 12-Dimethylbenz(a)-anthracene (DMBA)-induced apoptosis, and neither increased microRNA-214 (miR-214) expression nor DMBA treatment alone decreases NFKB1 expression in equine MDECs. a** Relative *NFKB1* expression in canine MDECs as compared to equine MDECs, detected by quantitative reverse transcription-polymerase chain reaction (qRT-PCR) using degenerate primers (**i**). Raw ct values of *NFKB1* and the reference gene *B2M* from qRT-PCR assays are presented, showing robust constitutive expression of *NFKB1* in both cell types (**ii**). **b** *NFKB1* expression in equine MDECs transfected with the miR-214 mimic or a negative miRNA mimic and treated with DMBA. **c** *NFKB1* expression in equine MDECs transfected with the miR-214 mimic or negative miRNA mimic. **d** *NFKB1* expression in equine MDECs treated with dimethylsulfoxide (DMSO) or DMBA. *n* = 3. Error bars show standard deviations. *P*-values are indicated on graphs.

independent of relatively low *NFKB1* expression controls the resistance of canine MDECs to DMBA-induced cell death. Although *NFKB1* expression varied between MDECs isolated from horses and dogs, expression was robust in each species based on the raw ct values detected by qRT-PCR (Fig. 6a(ii)). To confirm the accuracy of the degenerate PCR primers, we re-assessed *NFKB1* expression in DMBA-treated equine MDECs transfected with the miR-214 mimic and negative mimic. As expected, the results of decreased *NFKB1* expression in DMBA-treated equine MDECs transfected with the miR-214 mimic when compared to the negative mimic using the degenerative PCR primer set (Fig. 6b) mirrored the previously performed RT-PCR experiment using equine-specific primers (Fig. 3c), with the exception that the results in Fig. 6b reached significance whereas they were borderline non-significant in Fig. 3c.

To further assess the role of *NFKB1* expression in DMBA-induced apoptosis in equine MDECs, we evaluated *NFKB1* expression in equine MDECs that were either transfected with the miR-214 mimic in the absence of DMBA (Fig. 6c) or that were treated with DMBA in the absence of the miR-214 mimic

(Fig. 6d). Under both conditions, *NFKB1* expression did not change, indicating that increased levels of miR-214 and DMBA treatment are needed together to decrease *NFKB1* expression (Figs. 3c and 6b) and prevent apoptosis of equine MDECs.

## Discussion

We previously showed that mammosphere-derived epithelial cells (MDECs), formerly named mammary stem/progenitor cells (MaSCs), from mammals with a low mammary cancer incidence, such as equines, respond to 7, 12-Dimethylbenz(a)-anthracene (DMBA)-induced DNA damage by undergoing apoptosis[8]. In this study, we identified the microRNA miR-214-3p and its target gene nuclear factor kappa B subunit 1 (*NFKB1*) as key regulators of DMBA-induced apoptosis in equine MDECs. We also report on the direct interaction of miR-214-3p with the target gene unc-5 netrin receptor A (*UNC5A*).

MicroRNA-214 (miR-214) has been implicated in human breast cancer and proposed as a serum biomarker for malignant

disease and metastatic spread to regional lymph nodes[33–35]. However, very little has been published about the role of this miRNA in healthy mammary epithelial cells. Experiments in the murine epithelial cell line HC11 demonstrated that lactoferrin mRNA expression and protein synthesis is regulated by miR-214[24]. The same study also showed that inhibition of miR-214 expression leads to increased lactoferrin-induced apoptosis of the human mammary cancer cell line MCF7[24]. The latter supports the findings we made, namely that low levels of miR-214 can be correlated with enhanced carcinogen-induced apoptosis.

To explore which potential miR-214 target genes could be involved in this apoptotic response, we performed RNA sequencing analysis of miR-214 mimic-transfected, DMBA-treated equine MDECs which revealed 100 potential gene targets of miR-214-3p. We predicted *UNC5A* would be involved, based on (i) its role in mediating apoptosis[29,36,37]; (ii) the exceptionally strong miR-214-3p binding site of the 3' untranslated region (UTR) of *UNC5A*, consisting of 13 consecutive base pairings from the 5'-end of the miRNA; and (iii) the *UNC5A* ligand, netrin-1 *(NTN1)* gene being a proven target of miR-214-3p in bladder cancer cells[38]. Downregulation of *NTN1* was not detected in our RNA sequencing screen, suggesting that this ligand is not regulated by miR-214 in equine MDECs, and reinforcing the notion that miRNA activity varies according to species, cell type, and/or sub-cellular location[39]. While luciferase assays confirmed that miR-214 binds to the 3'UTR of *UNC5A*, knocking down this gene alone did not inhibit DMBA-induced apoptosis in equine MDECs. Regardless, the identification of *UNC5A* as a target of miR-214 contributes to the field of microRNA-mediated gene regulation.

The contributions of two additional predicted miR-214-3p target genes in DMBA-induced apoptosis of equine MDECs were examined, as their protein products promote apoptosis under specific circumstances. Mitogen-activated protein kinase 1 (MAPK1) is an enzyme that acts as a point of integration for a variety of biochemical signals and it controls many cellular processes, including the induction of apoptosis[40–43]. *MAPK1* has previously been reported to be a target of miR-214-3p in renal and hepatocellular carcinoma cells[44,45]. However, knocking down *MAPK1* alone in equine MDECs did not inhibit DMBA-induced apoptosis, suggesting that it is not sufficient to explain miR-214-3p-mediated regulation of apoptosis in DMBA-treated equine MDECs. The other predicted target gene of interest, *NFKB1*, encodes the transcription factor NF-κB. NF-κB is activated by various stimuli including cytokines, ultraviolet irradiation, and viral or bacterial products, leading to nuclear translocation and initiation of the expression of genes involved in a wide variety of biological processes, including apoptosis[46–49]. NF-κB is best known for its protective effects against programmed cell death, but it has been shown to be proapoptotic in response to specific stimuli in certain cell types[47,50]. Viruses including Dengue virus, Sindbis virus, and reoviruses promote apoptosis in infected cells, and this cell death is inhibited in cells lacking the NF-κB subunits p50 or p60, or by blocking NF-κB activity via protein decoys[51–53]. In vitro experiments using genetically modified tumor cell lines demonstrated that NF-κB is essential for p53-mediated cell death[54], and additional studies reported biochemical evidence that p53 may not necessarily stimulate the proapoptotic activity of NF-κB, but rather neutralizes its capacity to induce the expression of antiapoptotic genes[55,56]. Further experimental evidence suggests that NF-κB is not only differentially responsive depending on stimuli and cell type, but that the type of DNA damage (base modifications or strand breaks) determines if NF-κB modulates survival or apoptotic pathways[49], indicating that the role of NF-κB in apoptosis is highly context-dependent.

miR-214-3p has been shown to regulate NF-κB signaling in various contexts including bacteria-triggered inflammatory responses and osteoarthritis[57,58]. In the hg38 build of the human genome, the promoter region of miR-214 is predicted to occupy coordinates chr1:172,144,840–172,146,340[59]. Within this 1500 bp region, there are several proposed transcription factor binding sites including those for *FOXP2*, *SOX15*, and *RUNX1*, all of which are thought to work in opposition to NF-κB. In this study, increasing miR-214 or knocking down the miR-214 target *NFKB1* in equine MDECs inhibited DMBA-induced apoptosis, suggesting that miR-214-3p could modulate the apoptotic response of equine MDECs to DMBA through NF-κB signaling. This was further corroborated by the fact that the RNA sequencing data revealed that certain NF-κB-regulated genes are differentially expressed in miR-214 mimic-transfected, DMBA-treated equine MDECs (Supplementary Fig. 2).

How miR-214 exactly regulates NF-κB signaling in DMBA-treated equine MDECs remains to be determined, but the findings in this current study show that increasing miR-214 in combination with DMBA treatment results in a decrease in *NFKB1* expression, whereas each condition in itself does not. A potential mechanism could involve an endogenous miR-214 sponge, which is an RNA molecule carrying a miR-214 binding site that competitively binds miR-214, sequestering it and preventing it from binding its mRNA target[60], in this case, *NFKB1*. Interestingly, the cellular profiles of endogenous miRNA sponges in the form of circular and long non-coding RNAs have been shown to change in response to carcinogens[61,62], in some cases interacting with miRNAs altering apoptosis or susceptibility to cancer[63,64]. miRNAs and miRNA sponges can interact in positive feedback loops[65], which may be important in our model, and endogenous miRNA sponges that interact with miR-214 have been documented[45,66]. Based on our data combined with this knowledge, we propose that equine MDECs express robust levels of *NFKB1*, relatively low levels of miR-214, and some form of RNA that serves as a miR-214 sponge at baseline (Supplementary Fig. 5a). With the addition of a miR-214 mimic, a positive feedback loop leads to increased expression of the miR-214 sponge, which binds to and blocks the activity of the supplementary miR-214, and expression of *NFKB1* continues to be robust as miR-214 activity is blocked (Supplementary Fig. 5b). If equine MDECs are treated with DMBA alone, nothing changes from baseline, with cells expressing robust levels of *NFKB1*, relatively low levels of miR-214, and a form of RNA that serves as a miR-214 sponge (Supplementary Fig. 5c). However, when equine MDECs are transfected with a miR-214 mimic, increasing miR-214 levels in the cells, and then treated with DMBA, we propose that DMBA disrupts the miR-214-sponge positive feedback loop, leading to an excess of free miR-214 that can then bind to the 3' UTR of *NFKB1*, leading to decreased *NFKB1* expression (Supplementary Fig. 5d). This decrease in *NFKB1* prevents equine MDECs from undergoing apoptosis, leaving them vulnerable to the accumulation of oncogenic mutations when DMBA-induced DNA damage is repaired (Fig. 7).

Many molecules and pathways have the potential to be involved in the effect DMBA has on equine MDECs, and future experiments can be carried out to decipher these mechanisms and determine if they are conserved in MDECs isolated from other cancer-resistant species. The relationship between "high" miR-214-3p and "low" *NFKB1* does not appear to be involved in the resistance of canine MDECs to DMBA-induced cell death, as decreasing miR-214 did not lead to the decreased viability of DMBA-treated canine MDECs, and canine MDECs constitutively express more *NFKB1* than equine MDECs. Mechanisms that determine cancer resistance may not be related to those that contribute to cancer susceptibility.

Collectively, our data lead us to propose that relatively low levels of active miR-214 in equine MDECs allow for the robust expression of *NFKB1*, and apoptosis as a protective anti-cancer

**Fig. 7 Proposed mechanism by which microRNA-214 (miR-214) and nuclear factor kappa beta 1 (NFKB1) allow 7, 12-Dimethylbenz(a)anthracene (DMBA)-induced apoptosis in equine mammosphere-derived epithelial cells (MDECs). a** With DMBA treatment, relatively low levels of miR-214 allow for robust expression of *NFKB1*, which permits DMBA-induced apoptosis, protecting equine MDECs from oncogenic mutations that could arise if cells repair DMBA-induced DNA damage. **b** Addition of a miR-214 mimic prior to DMBA treatment leads to degradation of *NFKB1* and inhibition of *NFKB1*-mediated, DMBA-induced apoptosis, making equine MDECs potentially susceptible to oncogenic mutations that could arise if cells repair DMBA-induced DNA damage. **c** Introduction of *NFKB1* siRNA prior to DMBA treatment leads to a decrease in *NFKB1* expression and inhibition of *NFKB1*-mediated, DMBA-induced apoptosis, making equine MDECs potentially susceptible to oncogenic mutations that could arise if cells repair DMBA-induced DNA damage. Images created with BioRender.com.

response to the carcinogen DMBA (Fig. 7). Our experiments showed that either (i) increasing miR-214 by adding a miR-214 mimic by transfection or (ii) reducing *NFKB1* expression with *NFKB1*-specific siRNA prevents equine MDECs from undergoing apoptosis when treated with DMBA (Fig. 7).

## Methods

**Cell isolation, cells, and 7, 12-dimethylbenz(a)anthracene (DMBA) treatment**. Mammary gland tissues from euthanized healthy, non-lactating, research mares (4–20 years old) were harvested by removing 2 pieces of tissue, each 5 cm$^2$ from next to the median line of the second mammary gland compartments. Canine mammary gland samples were collected from euthanized healthy, non-lactating research beagles (6–10 years old), by removing a piece of tissue of at least 2 cm$^2$ from near the nipple. All animals were euthanized for reasons not related to this study. Mammary tissues were digested and processed to establish mammosphere-derived epithelial cell (MDEC) cultures, exactly as described before[67]. Briefly, tissues were minced into 0.3 cm$^3$ pieces before enzymatic digestion with 0.1% collagenase III (Worthington Biochemical Corporation, Lakewood, NJ) at 37 °C for 1 h. Digested tissues were passed through a 10 ml pipette several times, then filtered through 100 and 40 μm cell strainers (Corning, Glendale AZ) to obtain a single cell suspension. Cells were centrifuged at 300 x g, 5 min, room temperature to pellet, and resuspended in EpSC Medium consisting of 1:1 DMEM: Ham's F12 (Corning), 10% fetal bovine serum (FBS) (R&D Systems, Minneapolis, MN), B27 without vitamin A (Life Technologies, Grand Island, NY), antibiotic-antimycotic solution (Corning), 10 ng/ml human FGF-2 (Sigma Aldrich), and 10 ng/ml human EGF (Sigma Aldrich). Cells were plated in 6-well tissue culture plate wells (Corning) for 1 h, during which time adherent cells attached to the plate. Cells in suspension were passed into clean 6-well tissue culture plate wells for an additional hour to remove more adherent cells. Cells in suspension were passed into

ultralow attachment plates (Corning) where they formed mammospheres, enriched in progenitor cells. Cells remained in mammospheres for about two weeks, after which time they were dissociated, plated on cell culture plastic, and maintained as adherent cultures in EpSC Medium. MDECs were plated at densities of $3 \times 10^4$ cells per well in 48-well culture plates for MTT assays, $6 \times 10^4$ cells per well in 24-well culture plates for anti-active caspase-3 antibody binding, $12 \times 10^4$ cells per well in 12-well culture plates for RNA isolation, and $8 \times 10^4$ cells per well in 24-well culture plates for luciferase assays.

MDECs were treated with 5 μM DMBA diluted in dimethyl sulfoxide (DMSO) or an equivalent volume of DMSO (vehicle control) in EpSC Medium 48 h post plating.

HeLa cells were cultured in DMEM with 10% FBS, and penicillin-streptomycin (Corning) and plated at a density of $8 \times 10^4$ cells per well in 24-well culture plates for luciferase assays.

**miRNA isolation and qRT-PCR**. Total RNA for miRNA analysis was isolated from snap-frozen MDEC pellets using a total RNA purification kit, according to the manufacturer's instructions (Norgen Biotek, Ontario, Canada). RNA quantity was measured using a NanoDrop spectrophotometer (Thermo Fisher). miRNA was reverse transcribed and amplified using TaqMan® MicroRNA Assay reagents, according to the manufacturer's instructions (Thermo Fisher). The reverse transcription reaction was performed in a MasterCycler thermocycler (Eppendorf, Hauppauge, NY) and PCR was carried out in an ABI QuantStudio3 thermocycler (Thermo Fisher).

**miRNA-214 knock-down and overexpression**. At 24 h post-plating, when cells were approximately 85% confluent, Lipofectamine 3000 reagents (Thermo Fisher) were used to transfect MDEC with *mir*Vana™ miR-214 or negative mimic, Anti-miR™ miR-214 inhibitor or negative inhibitor (Thermo Fisher), according to manufacturer's instructions. Cells were analyzed 48 h post transfection.

**Viability assays**. After treating MDECs for 24 h with DMBA, a 3-(4,5-dimethylthiazol-2-yl)-2,5-diphenyltetrazolium bromide (MTT) in vitro toxicology assay (Sigma Aldrich) was carried out according to manufacturer's recommendations, as previously described[68]. Absorbances were measured at 570 nm and 690 nm on a Tecan Infinite 200 PRO plate reader (Tecan, Morrisville, NC). Each assay contained cells from three individual MDEC lines (biological replicates) and each condition was run in three technical replicates per assay. For primary experiments (assays comparing the responses of canine versus equine MDECs to DMBA and assays comparing the responses of equine MDECs transfected with the miR-214 mimic versus controls to DMBA) assays were carried out three times.

Caspase activity assays were performed by using TF2-VAD-FMK as a fluorescent indicator for activated caspase-1, -3, -4, -5, -6, -7, -8, and -9 in apoptotic cells, according to the manufacturer's instructions (Abcam, Cambridge, MA). Fluorescence was measured using a BD Fortessa X-20 flow cytometer (BD Biosciences, San Jose, CA) and data were analyzed using FlowJo software (Ashland, OR). Prior to conducting assays, brightfield images of cell cultures were captured using an Olympus BX51 light microscope (Olympus, Tokyo, Japan). Each assay contained cells from three individual MDEC lines (biological replicates).

Active caspase-3 immunostaining was performed, as previously described[69]. Briefly, cells were fixed with 4% paraformaldehyde (PFA), washed with phosphate-buffered saline (PBS), and permeabilized with 0.5% Triton X-100 (Sigma Aldrich) for 10 min. Following a 30-min incubation in blocking solution (1% goat serum and 1% bovine serum albumin in PBS) at room temperature, fixed cultures were probed with a rabbit anti-active caspase-3 antibody (Abcam) or rabbit IgG (Table 1), and incubated overnight at 4 °C. Cells were washed with PBS and an HRP-conjugated goat anti-rabbit secondary antibody (Jackson ImmunoResearch Labs, West Grove, PA), diluted 1:100 in PBS, was added. After 30 min at room temperature, cells were washed with PBS and AEC solution (Invitrogen Life Technologies) was added for 15 min. Cells were counterstained with Gill's Hematoxylin (Thermo Fisher) and washed with tap water. Images of labeled cells were captured with a digital camera mounted on an Olympus BX51 light microscope (Olympus). To determine the percentage of cells positive for active caspase-3,

cells in five fields were counted and classified as either positive or negative based on the presence or absence of red staining. Immunostaining was performed with three individual MDEC lines (biological replicates) and each condition was run in three technical replicates.

**RNA sequencing**. Total RNA for sequencing was extracted with TRIzol reagent, as described previously[70], and RNA quantity was measured using NanoDrop spectrophotometer (Thermo Fisher). The quality of total RNA was evaluated using an Agilent TapeStation (Agilent). RNA-seq libraries were prepared with the NEBNext Ultra Directional RNA Library Prep Kit (NEB, Ipswich, MA) using 500 ng total RNA followed by polyA + enrichment and were sequenced using Illumina NextSeq500 to obtain 81 nucleotide single-end reads. Sequencing reads were aligned to the *Equus caballus* reference genome (r2.9) using the STAR aligner (v2.4.2) and reads aligning to the transcriptome were quantified using Salmon (v0.8.2). An average of 24.5 million reads mapped per sample at an average mapping rate of 73.2%. Differential expression was determined using DESeq2 (v1.3) as previously described[22].

**RNA isolation and qRT-PCR**. Total RNA for mRNA analysis was isolated from snap-frozen MDEC pellets using a total RNA purification kit, according to the manufacturer's instructions (QIAGEN, Germantown, MD). RNA quantity was measured using a NanoDrop spectrophotometer (Thermo Fisher). SYBR green–based RT-PCR was performed to determine fold-change in transcripts of equine mitogen-activated protein kinase 1 (*MAPK1*), nuclear factor kappa beta 1 (*NFKB1*), and unc-5 netrin receptor A (*UNC5A*), and data were analyzed, as previously described[71,72]. A MasterCycler thermocycler (Eppendorf) was used for the reverse transcription reaction and PCR was carried out in an ABI QuantStudio3 thermocycler (Thermo Fisher). The previously validated reference gene equine beta-2-microglobulin (*B2M*) was used to normalize samples[68], and all samples were run in triplicate. Primer sequences and gene IDs are listed in Table 2.

**Luciferase assays**. At 24 h post-plating, when cells were approximately 85% confluent, cells were transfected with a miRNA 3'UTR target expression clone for human *UNC5A* (GeneCopoeia, Rockville, MD) and *mir*Vana™ miR-214 mimic or Anti-miR™ miR-214 inhibitor using Lipofectamine 3000 reagents, according to manufacturer's instructions. After 48 h, cells were collected and the Luc-Pair Duo-Luciferase Assay Kit 2.0 (Gene-Copoeia) was used to determine the extent of miR-214 binding to the 3'UTR of *UNC5A*. Luminescence was detected with an Infinite 200 Pro plate reader (Tecan). To confirm the binding of miR-214 to the 3'UTR of *UNC5A*, an expression clone was generated with a 3-nucleotide deletion in the predicted miR-214 binding site, and assays were repeated with this mutant clone. Each condition was run in three technical replicates per assay, each replicate was run in duplicate wells.

---

**Table 1 Primary Antibodies.**

| Target | Host | Clone | Dilution | Manufacturer |
|---|---|---|---|---|
| Active caspase-3 | Rabbit | N/A | 1:100 | Abcam (ab4051) |
| UNC5A | Rabbit | N/A | 1:200 | LS Bio (C80729) |
| MAPK1 | Rabbit | N/A | 1:200 | LS Bio (C117080) |
| NF-κB | Rabbit | N/A | 1:200 | LS Bio (B8072) |
| β-actin | Rabbit | N/A | 1:1000 | Abcam (ab8227) |
| Rabbit IgG | Rabbit | EPR25A | 1:100 | Abcam (ab172730) |

*N/A* not applicable

---

**Table 2 Primers used for RT-PCR.**

| Full Name | Abbreviation | Gene ID | Forward primer (5'-3') | Reverse primer (5'-3') |
|---|---|---|---|---|
| Mitogen-activated protein kinase 1 | *MAPK1* | 100057701 | TGACCTCAAACCTTCCAACC | TTCTGGAGCCCTGTACCAAC |
| Nuclear factor kappa B subunit 1 | *NFKB1* | 100067894 | CATCTTTGACAACCGCGCC | GCCTGGTCCCGTGAAATACA |
| Nuclear factor kappa B subunit 1 (degenerate) | *NFKB1* | 100067894/ 442859 | CACCTAGCTGCCAAAGAAGG | CATCAYGGCTATGTGAATGG |
| Unc-5 netrin receptor A | *UNC5A* | 100068742 | AGAGCAAGCTTCTCGTCAGC | GTGAAGGTGCAGTGCAGGTA |
| Beta-2-microglobulin | *B2M* | 100034203 | TCTTTCAGCAAGGACTGGTCTTT | CATCCACACCATTGGGAGTAAA |

**Small-interfering RNA (siRNA) knockdown**. Silencer Select siRNAs targeting equine *MAPK1, NFKB1,* and *UNC5A*, were designed by Thermo Fisher. A BLAST search (https://blast.ncbi. nlm.nih. gov/Blast.cgi) confirmed that Silencer Select Negative Control #2 siRNA (Thermo Fisher) is not complementary to any equine genes, so it was used as a non-specific (negative) control. siRNA knockdown was carried out according to the manufacturer's recommendations, as described previously[73]. Each assay contained cells from three individual MDEC lines (biological replicates) and each condition was run in three technical replicates per assay.

**Western blot analysis**. Cells were collected, protein concentrations were determined, and Western blotting was used to determine the efficacy of UNC5A, MAPK1, and NF-κB silencing in equine MDECs by siRNA, exactly as previously described[68]. Antibodies used in Western blots are shown in Table 1. Gel imaging and determination of band density relative to β-actin loading controls were also carried out as previously described[68].

**Statistics and reproducibility**. The student's *t*-test for unpaired data was used to test for statistically significant differences in gene expression where two conditions were compared. One-way ANOVA, followed by Tukey's multiple comparison test was used to determine statistically significant differences in gene expression where more than two conditions were compared, i.e., cell viability, number of active caspase-3 positive cells per field, luciferase activity, and Western blot band density. GraphPad software was used for analysis (GraphPad, La Jolla, CA). Each experiment consisted of three biological replicates (MDEC cell lines from 3 different animals), that were run in duplicate or triplicate, depending on the assay. Primary experiments (MTT assays comparing the responses of canine versus equine MDECs to DMBA and MTT assays comparing the responses of equine MDECs transfected with the miR-214 mimic versus controls to DMBA) were repeated 3 times. Data shown are the means of biological replicates or the means of three experiments, when applicable. The bars show standard deviations. $P < 0.05$.

**Reporting summary**. Further information on research design is available in the Nature Portfolio Reporting Summary linked to this article.

## Data availability

The miRNA sequencing data are accessible through GEOSeries #GSE126424. The RNA sequencing data are accessible through GEOSeries #GSE189879. Source data can be found in Supplementary Data 3. All other data are available from the authors on reasonable request.

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

## Acknowledgements

We would like to acknowledge Dr. Jen Grenier from the Cornell Transcriptional Regulation & Expression Facility for her excellent technical assistance. This work was supported by unrestricted funding from the Albert C. Bostwick Foundation to G. Van de Walle. This funding source had no role in the study design, data collection, data analysis, data interpretation, writing of the report, or the decision to submit this manuscript for publication.

## Author contributions

R.M.H. collected, assembled, analyzed and interpreted data, and wrote the manuscript. S.P.D. collected, assembled and analyzed data, and contributed to the manuscript. M.K. assembled and analyzed data, and edited the manuscript. P.S. designed experiments, interpreted data, and edited the manuscript. G.R.V.d.W. conceptualized experiments, interpreted data, and contributed to manuscript writing. All authors approved the final manuscript.

## Competing interests

The authors declare no competing interests.
