## [Peer Review File · Communications Biology]

Reviewers' comments:

Reviewer #1 (Remarks to the Author):

In this manuscript entitled – Low miRNA-214-3p stimulates carcinogen-induced mammary epithelial cell apoptosis in mammary cancer-resistance species – the authors probed their previously published data set to identify 53 miRNAs differentially expressed between canine and equine MDECs. In my opinion, research exploring the molecular mechanisms that underlie species-specific differences in cancer occurrence could lead to impactful findings. Here, the authors selected 4 miRNAs for further study because of their known involvement in apoptosis. Of these, one was selected (miR-214) for in depth study based on its overall higher expression in MDECs. Overexpression of miR-214 showed rescue of a previously identified difference in response of equine and canine MaSCs to DMBA treatment --- that is miR-214 overexpression inhibited the apoptosis that equine MDECs undergo in response to DMBA. According to the authors' model, apoptosis is a good thing in terms of cancer and it may explain why horses are relatively less burdened by mammary cancers than dogs. Further sequencing of equine MDECs transfected, with miR-214 mimic or negative mimic, + DMBA treatment showed 100 genes predicted to be miR-214 targets. These were prioritized based on a role in apoptosis and whether their "target" genes were downregulated; 3 genes apparently fit this bill: MAPK1, UNC5A and NFKB1. Before knocking down these genes to analyze their role in DMBA-induced viability, the authors evaluated the binding of miR-214 to Unc5A, a previously unidentified target of miR-214, using luciferase assays. Then, they collectively knocked down the genes before selectively knocking down each gene. Only knockdown of NFKB1 inhibited DMBA-induced loss of viability (apoptosis was not assayed). But the mechanism underlying this action was not explored.

My overall impression is that the research is solid (and appropriately analyzed statistically); the discoveries are potentially impactful. Certainly, this paper, albeit trim, sets up interesting questions. How, in equine MDECs, does NFKB1 cause DMBA-induced apoptosis/loss of viability or, asked another way, how does knockdown of NFKB1 inhibit DMBA-induced apoptosis/loss of viability (please note the authors conflate loss of viability and apoptosis). Unfortunately, it does not appear that the answer concerning NFKB1 regulation will inform the situation in canine MaSCs/MDECs, which do not undergo apoptosis, which the authors posited make them more susceptible to cancer, because reducing miR-214 in canine MDECs did not cause decreased viability upon DMBA treatment. Considering how the manuscript was framed, this does reduce the overall impact. In any case, in my opinion, the manuscript should be strengthened by additional data that would augment the case being made about NFKB1 in equine MDECs. 1) The authors base conclusions about MAPK1, UNC5A and NFKB1 knockdown without verifying that protein levels were decreased. They need to perform immunoblot to verify knockdown. 2) The authors conflate viability and apoptosis using a viability assay to make conclusions about apoptosis. They need to use appropriate assays and measure apoptosis in order to draw their conclusions. This is true in a number of places in the manuscript including Sup Fig 1c and Figure 5. 3) It would greatly strengthen the manuscript if the authors provide some insight into how NFKB1 makes cells undergo DMBA-induced apoptosis or, alternatively, how downregulation of NFKB1 prevents DMBA-induced apoptosis of equine MDECs (and for thoroughness they should examine the consequences of NFKB1 overexpression on DMBA-induced apoptosis in canine MaSCs/MDECs). Figure 4 on miR-214 could go into the supplement because, while it yields a clear result, the result does not matter for this manuscript because even though miR-214 binds Unc5A, knockdown of UNC5A (please show this occurs -see above) does not rescue viability of DMBA-treated equine MDECs.

Below are specific comments on the figures.

Figure 1 shows the effects of DMBA treatment on miRs in horse or dog MaSCs. Having read previous studies from this lab (Ledet et al., which is also reference in the first sentence of the results, I am confused about the difference between mammosphere-derived epithelial cell (MDEC) cultures studied here and the mammary stem/progenitor cells (MaSC) studies previously. The authors should clarify why they used MDEC cultures here and how this differs from the MaSC cultures described in Ledet et

al. Otherwise, the data presented in this figure are straightforward, showing the differential expression of miRs in canine and equine MDEC with the authors choosing to study miR-214, which is lower (~50% reduced) in equine MDEC, due to its higher overall level of expression in cultures from both dog and horse. Please note that this is a key point; they chose miR-214 because of its robust levels of expression in both canine and equine MDECs. The conundrum is this sentence from the abstract, "We propose that low levels of miR-214 allow NFKB1 expression and apoptosis in response to DMBA in MDECs from mammals with low mammary cancer incidence." But, miR-214 isn't particularly expressed at low levels in equine MDECs.....so how does work?

The authors previously showed that equine DMBA-damaged MDEC are eliminated by apoptosis so their viability decreases upon DMBA treatment. Figure 2 shows how increasing miR-214 expression in equine MDECs resulted in no significant effect on their cell viability after treatment with DMBA when compared to controls (i.e. increasing miR-214 rescued the equine effect). The authors should show the caspase-3 immunostaining. Also in panel 2bii the phase contrast images aren't representative of the quantified data. Images of control DMSO treatment look very different from No Treatment but this is not the case according to the quantified % viability. Also, DMBA-treated miR-214 mimic cells look different from No Treatment and DMSO but according to the quantified viability assay there should be no difference. If the viability assays were performed on cells at vastly different cell densities, it calls the results into question. This should be carefully considered and corrected by the authors.

The authors should briefly describe the ThermoFisher technology they used to manipulate miR expression more clearly in their Results section.

Figure 3 identifies 100 genes predicted to be miR-214 targets. These were prioritized based on a role in apoptosis and whether their "target" genes were downregulated; 3 genes apparently fit this bill: MAPK1, UNC5A and NFKB1. Maybe I misunderstood this point, "Predicted target genes were prioritized based on (ii) an observed decrease in expression of genes downstream of the predicted target in the RNA-seq analysis, suggesting miRNA regulation of a universal upstream regulator." But, I am confused as to why it matters if the target genes of the genes downregulated by miR-214 are downregulated. That is to say, downregulation of a target could result in either up or down regulation of gene expression; the targets of miR-214-downregulated targets aren't necessarily downregulated---are they? In any case, it would be appropriate for the authors to show in a Supplementary figure the downstream target genes of these candidates whose expression was decreased i.e. the targets that allowed the candidates to be identified as such. I note that the authors offer up two references (Perrot-Applanat et al and Wu et al.) Both appear to be about NFKB1; neither appear to be about UNC5A or MAPK1.

Figure 4 identifies a strong potential miR-214-3p binding site and since UNC5A has not previously been validated as a target of miR-214, the authors perform luciferase assays in HeLa cells (as a proxy for equine MDECs) because they both express miR-214. I need to point out that both canine and equine MDECs express miR-214 but they apparently act very differently to miR-214's actions so I am not sure another cell line acts as an appropriate proxy. But I guess the point is that you can use HeLa cells, which express miR-214 to assay if this miR binds a target. Transfecting in the 3'UTR of UNC5A under control of the SV40 promoter, the authors find that FLuc/RLuc significantly decreased in the presence of the miR-214 mimic and significantly increased in the presence of the miR-214 inhibitor – evidence of binding. They also used UNC5A 3'UTR with a mutation that led to the signal being abolished. This is a fine set of studies, but really beside the point because Figure 5 will show that UNC5A is not a relevant target for miR-214 in regulating DMBA-induced loss of viability.

Figure 5 shows the knockdown experiments. At first, all 3 genes are knocked down in a cocktail and then they are knocked down individually. (A) show viability in control versus triple KD. I couldn't help but notice that the decrease in viability in the controls of this experiment is quite a bit less (25% or so) than in other experiments (e.g. Figure 2b and Figure 5b compared to Figure 5a, c, d). In fact, it looks as if the control and DMBA treatment for Figure 5a, c, d are identical, which is odd and should

be checked. Then the genes are knocked down individually and only NFKB1 knockdown results in a rescue of the reduction in viability in DMBA treated equine MDEC samples. Figure 5 is underwhelming and should be improved. RT-qPCR is not an effective way to determine whether RNA interference reduced the expression levels of these target proteins; immunoblots need to be done to confirm the knockdown. Assays that measure apoptosis rather than simply cell viability should be performed and phase contrast pictures of these cells should be shown to the reader. As a control, it would be appropriate to manipulate the expression of NFKB1 in canine MDECs to see if there is an effect on apoptosis. Moreover, there should be some insight into the mechanism of NFKB1 action in equine MDECs. Does DMBA treatment regulate NFKB expression in MDECs? Is NF- κ B or any of the candidate NF- κ B-regulated proteins -- cyclin F (CCNF), cyclin E2 (CCNE2), colony stimulating factor 1 (CSF1) or the BRCA1-associated RING domain 1 (BARD1) involved in the DMBA-induced apoptotic response? And circling back to miR-214, where is this miR encoded in the genome and what are the regulatory mechanisms controlling its expression?

Minor pt. The figure is labelled Scramble siRNA but the manuscript refers to negative siRNA-transfected. It would be less confusing to keep the nomenclature similar between figure and manuscript.

Reviewer #2 (Remarks to the Author):

In this paper, the authors investigated miR-214-3p's role in DMBA-induced apoptosis of equine MDECs and observed that low levels of miR-214 promote NFKB1 expression and thereby apoptosis in response to DMBA in equine MDECs. Further, it describes unc-5 netrin receptor A (UNC5A) as a downstream target of miR-214-3p for the first time. However, the conclusions drawn from the study are exaggerated and there are certain points that need to be addressed.

1. The role of NFKB in apoptosis should be clearly described.
2. Cell viability study is insufficient to prove the DMBA-induced apoptosis in equine MDECs. Flow cytometry study should be conducted to confirm it.
3. Cell viability is insufficient to establish the involvement of NFKB.
4. Units like micromolar (μ M) and micrometre (μ m) should be corrected.
5. Objective and significance of this study is not clear. Describe it properly. Further, the paper should be written in a clear and concise way.

We would like to thank the editor and both reviewers for their thorough reading and constructive suggestions. We found the comments useful in guiding our substantial revisions and we believe the quality of the manuscript has greatly improved. It is our hope that the revised manuscript is acceptable for publication in Communications Biology.

Comments from the Editor:

Your manuscript entitled "Low miRNA-214-3p stimulates carcinogen-induced mammary epithelial cell apoptosis in mammary cancer-resistance species" has now been seen by 2 referees, whose comments are appended below. You will see from their comments copied below that while they find your work of potential interest, they have raised quite substantial concerns that must be addressed. Should further experimental data or analysis allow you to address these criticisms, we would be happy to look at a substantially revised manuscript. In particular, please note that the following revisions would be necessary for us to contact our referees again: We would like you to make a stronger case for the role of NFKB1 in the horse mammosphere-derived epithelial cells (MDECs). Assays should be included to show that the reduction in viability is actually apoptosis. For the knockdown experiments in Figure 5 we want you to include effects on protein level.

>> First of all, we thank the editor for their encouraging words related to our work and for giving us the opportunity to resubmit a substantially revised manuscript.

>> Specifically, we responded to the necessary revisions as follows:

(1) In response to the comment about making a stronger case for the role of NFKB1 in the horse MDECs, we performed additional experiments which are described in detail in our response to reviewer 1, comment 15. Moreover, we added additional text describing the role of NFKB1 in apoptosis as requested by reviewer 2, comment 1.

(2) We have now also included assays to show that the reduction in viability is indeed through apoptosis. These additional experiments are described in more detail in our response to reviewer 1, comment 14, and reviewer 2, comment 2.

(3) We have now also performed immunoblots to confirm that siRNA targets were reduced at the protein level as well and these additional experiments are described in more detail in our response to reviewer 1, comment 13.

>> In addition to these necessary revisions, we have also responded to all other issues raised by the reviewers, as outlined in our point-by-point responses below.

Comments from Reviewer #1:

General comments:

My overall impression is that the research is solid (and appropriately analyzed statistically); the discoveries are potentially impactful. Certainly, this paper, albeit trim, sets up interesting questions. How, in equine MDECs, does NFKB1 cause DMBA-induced apoptosis/loss of viability or, asked another way, how does knockdown of NFKB1 inhibit DMBA-induced apoptosis/loss of viability (please note the authors conflate loss of viability and apoptosis). Unfortunately, it does not appear that the answer concerning NFKB1 regulation will inform the situation in canine MaSCs/MDECs, which do not undergo apoptosis, which the authors posited make them more susceptible to cancer, because reducing miR-214 in canine MDECs did not cause decreased viability upon DMBA treatment. Considering how the manuscript was framed, this does reduce the overall impact. In any case, in my opinion, the manuscript should be strengthened by additional data that would augment the case being made about NFKB1 in equine MDECs. 1) The authors base conclusions about MAPK1, UNC5A and NFKB1 knockdown without verifying that protein levels were decreased. They need to perform immunoblot to verify knockdown. 2) The

authors conflate viability and apoptosis using a viability assay to make conclusions about apoptosis. They need to use appropriate assays and measure apoptosis in order to draw their conclusions. This is true in a number of places in the manuscript including Sup Fig 1c and Figure 5. 3) It would greatly strengthen the manuscript if the authors provide some insight into how NFkB1 makes cells undergo DMBA-induced apoptosis or, alternatively, how downregulation of NFkB1 prevents DMBA-induced apoptosis of equine MDECs (and for thoroughness they should examine the consequences of NFkB1 overexpression on DMBA-induced apoptosis in canine MaSCs/MDECs). Figure 4 on miR-214 could go into the supplement because, while it yields a clear result, the result does not matter for this manuscript because even though miR-214 binds Unc5A, knockdown of UNC5A (please show this occurs -see above) does not rescue viability of DMBA-treated equine MDECs.

>> *We thank the reviewer for their encouraging words related to our solid research and its potential impactful nature. We fully agree with the proposed suggestions to strengthen our paper, and have addressed those accordingly, as described under the specific comments.*

Specific comments:

1. Figure 1 shows the effects of DMBA treatment on miRs in horse or dog MaSCs. Having read previous studies from this lab (Ledet et al., which is also reference in the first sentence of the results, I am confused about the difference between mammosphere-derived epithelial cell (MDEC) cultures studied here and the mammary stem/progenitor cells (MaSC) studies previously. The authors should clarify why they used MDEC cultures here and how this differs from the MaSC cultures described in Ledet et al.

>> *Thank you for bringing this to our attention. After the Ledet work was published in 2018, we re-named the MaSCs to describe the cells more accurately based on how they are generated. The renamed mammosphere-derived epithelial cells (MDECs) are isolated and cultured exactly as they were for our previous work. To avoid this confusion, we modified the text to include the acronym MaSC [Page 3, lines 51, 55, 57 & 58; Page 4, line 78; and Page 12, lines 256-257], and we provide an explanation of how and why we renamed the cells MDECs [Pages 4, lines 79-82].*

2. Otherwise, the data presented in this figure are straightforward, showing the differential expression of miRs in canine and equine MDEC with the authors choosing to study miR-214, which is lower (~50% reduced) in equine MDEC, due to its higher overall level of expression in cultures from both dog and horse. Please note that this is a key point; they chose miR-214 because of its robust levels of expression in both canine and equine MDECs. The conundrum is this sentence from the abstract, "We propose that low levels of miR-214 allow NFkB1 expression and apoptosis in response to DMBA in MDECs from mammals with low mammary cancer incidence." But, miR-214 isn't particularly expressed at low levels in equine MDECs.....so how does work?

>> *We removed the word "low" from the title and added the word "relatively" to the abstract [Page 2, line 33] to clarify that our focus on miR-214 is based on the fact that it is expressed at a lower level in a mammal with low mammary cancer incidence (i.e. horse) compared to a mammal with high incidence of mammary cancer (i.e. dog).*

3. The authors previously showed that equine DMBA-damaged MDECs are eliminated by apoptosis so their viability decreases upon DMBA treatment. Figure 2 shows how increasing miR-214 expression in equine MDECs resulted in no significant effect on their cell viability after treatment with DMBA when compared to controls (i.e. increasing miR-214 rescued the equine effect). The authors should show the caspase-3 immunostaining.

>> *As requested, we now show the caspase-3 immunostaining in Figure 2c, with quantification in the 2c(i) panel and representative images in the 2c(ii) panel. We also added arrows to the 2c(ii) panel that point to red (positive) cells. Also, and in response to a comment from reviewer #2, we*

performed flow cytometry experiments to detect caspase activity in equine MDECs transfected with the miR-214 mimic or negative mimic and treated with DMBA. These new data are shown in Figure 2b and are described in the Results section [Page 7, lines 130-135]. The figure legend has been adjusted accordingly [Page 31, lines 676-679 and Page 31, lines 683-684].

Figure 2. Harman et al.

4. Also in panel 2bii the phase contrast images aren't representative of the quantified data. Images of control DMSO treatment look very different from No Treatment but this is not the case according

to the quantified % viability. Also, DMBA-treated miR-214 mimic cells look different from No Treatment and DMSO but according to the quantified viability assay there should be no difference. If the viability assays were performed on cells at vastly different cell densities, it calls the results into question. This should be carefully considered and corrected by the authors.

>> *We apologize for the confusion. The original 2b(ii) panel showed representative images of the cells after caspase-3 antibody binding, not representative images of the quantified viability showed in the original 2b(i) panel. To fix this, we added a modified panel 2a(ii) to the figure, showing representative phase contrast images of the cells before the viability assays were carried out, which are shown quantitatively in panel 2a(i). We also added arrows to the 2a(ii) panel that point to dead cells. The results presented in Figure 2 are now described in more detail in the Results section [Pages 6-7, lines 129-135] and the legend has been adjusted accordingly [Page 31, line 676 and Page 31, lines 683-684].*

5. The authors should briefly describe the ThermoFisher technology they used to manipulate miR expression more clearly in their Results section.

>> *As requested, we added additional text to briefly describe the mimic and inhibitor technology used to manipulate miR expression in the Results section [Page 6, lines 111-113 and Page 7, lines 141-143].*

6. Figure 3 identifies 100 genes predicted to be miR-214 targets. These were prioritized based on a role in apoptosis and whether their “target” genes were downregulated; 3 genes apparently fit this bill: MAPK1, UNC5A and NFKB1. Maybe I misunderstood this point, “Predicted target genes were prioritized based on (ii) an observed decrease in expression of genes downstream of the predicted target in the RNA-seq analysis, suggesting miRNA regulation of a universal upstream regulator.” But, I am confused as to why it matters if the target genes of the genes downregulated by miR-214 are downregulated. That is to say, downregulation of a target could result in either up or down regulation of gene expression; the targets of miR-214-downregulated targets aren’t necessarily downregulated---are they?

>> *Thank you for pointing this out. Direct target genes of miR-214 should be downregulated, but the reviewer is correct in that genes downstream of target genes can either be up- or downregulated. We have replaced the word “decrease” by “change” in the sentence [Page 8, lines 163-164].*

7. In any case, it would be appropriate for the authors to show in a Supplementary figure the downstream target gene of these candidates whose expression was decreased i.e. the targets that allowed the candidates to be identified as such.

>> *As requested, we included a Supplementary Figure (Supplemental Figure 2) showing the 3 candidate target genes and select downstream genes that were found to be up- or downregulated in the RNA-seq analysis. This new Supplemental figure is referred to in the Results section [Page 8, line 168] and a legend for it has been created [legend included with figure].*

8. I note that the authors offer up two references (Perrot-Appianat et al and Wu et al.). Both appear to be about NFKB1; neither appear to be about UNC5A or MAPK1.

>> *We thank the reviewer for catching this and we have now included the appropriate references for all three genes [Page 8, lines 166-168].*

9. Figure 4 identifies a strong potential miR-214-3p binding site and since UNC5A has not previously been validated as a target of miR-214, the authors perform luciferase assays in Hela cells (as a proxy for equine MDECs) because they both express miR-214. I need to point out that both canine and equine MDECs express miR-214 but they apparently act very differently to miR-

214's actions so I am not sure another cell line acts as an appropriate proxy. But I guess the point is that you can use Hela cells, which express miR-214 to assay if this miR binds a target. Transfecting in the 3'UTR of UNC5A under control of the SV40 promotor, the authors find that FLuc/RLuc significantly decreased in the presence of the miR-214 mimic and significantly increased in the presence of the miR-214 inhibitor – evidence of binding. They also used UNC5A 3'UTR with a mutation that led to the signal being abolished. This is a fine set of studies, but really beside the point because Figure 5 will show that UNC5A is not a relevant target for miR-214 in regulating DMBA-induced loss of viability.

>> We appreciate the kind words indicating our UNC5A experiments are a fine set of studies, and although we agree that it loses some impact as we later find out that UNC5A is not a relevant target for miR-214 in regulating DMBA-induced loss of viability, we decided to present these data as a main figure based on the novelty of identifying this gene as a miR-214 target for the first time.

Supplemental Figure 2. Transfection with a microRNA-214 (miR-214)-mimic and treatment with 7, 12- Dimethylbenz(a)anthracene (DMBA) leads to downregulation of unc-5 netrin receptor A (UNC5A), mitogen-activated protein kinase 1 (MAPK1), and nuclear factor kappa beta 1 (NFKB1) in equine mammosphere-derived epithelial cells (MDECs). Up and downregulation of some downstream targets of these 3 transcripts are listed.

10. Figure 5 shows the knockdown experiments. At first, all 3 genes are knocked down in a cocktail and then they are knocked down individually. (A) show viability in control versus triple KD. I couldn't help but notice that the decrease in viability in the controls of this experiment is quite a bit less (25% or so) than in other experiments (e.g. Figure 2b and Figure 5b compared to Figure 5a, c, d).

>> *We fully agree with the reviewer that the decrease in viability can vary across experiments, as we have noticed this as well. We attribute that to working with primary cell cultures that are at different passages at different points in time when these experiments are carried out. Moreover, we do want to point out that all these experiments are carried out using equine MDECs isolated from 3 different animals. So in addition to performing 3 technical replicates for each experiment, each experiment is done with 3 biological replicates, which can introduce variability in the degree of DMBA-induced killing across experiments. Still, and despite the variability in decreased viability across experiments, the decrease is always statistically significant.*

11. In fact, it looks as if the control and DMBA treatment for Figure 5a, c, d are identical, which is odd and should be checked.

>> *The reviewer is correct in pointing out that the control and negative siRNA data in Figure 5c(i) and 5d(i) are the same. This is because we tested the MAPK1 and NFKB1 siRNAs in the same assay in which we used one set of control and negative siRNA culture wells. We decided to present these control and negative siRNA data twice for ease of interpretation and visual appeal, but we agree that in doing so, clarification is needed. Therefore, we have now included this disclaimer in the legend of Figure 5 [Page 32, lines 718-719]. Although the control and negative siRNA data in 5a(i) look similar to 5c(i) and 5d(i), they were generated in different assays.*

12. Then the genes are knocked down individually and only NFKB1 knockdown results in a rescue of the reduction in viability in DMBA treated equine MDEC samples. Figure 5 is underwhelming and should be improved.

>> *Figure 5 has been improved by including additional flow cytometry experiments to assess apoptosis in addition to cell viability ((ii) panels), in line with this reviewer's comment 14 and reviewer #2's comment 2. The flow cytometry procedure has been added to the Materials and Methods section [Page 17, lines 370-375]. The new data in Figure 5 are referred to in the Results section [Page 10, lines 215-216 and Page 11, lines 223-226] and the legend has been adjusted accordingly [Page 32, lines 708-717].*

Figure 5. Harman et al.

13. RT-qPCR is not an effective way to determine whether RNA interference reduced the expression levels of these target proteins; immunoblots need to be done to confirm the knockdown.

>> We fully agree with the reviewer's comment and as requested; we have performed immunoblots to confirm the siRNA knockdowns at the protein level as well. Data are shown in Supplemental Figure 3, panels a (triple knockdown), b (UNC5A knockdown), c (MAPK1 knockdown), and d (NFKB1 knockdown). Primary antibodies used are listed in Table 1 and the Western blot procedure has been added to the Materials and Methods section [Page 20, lines 436-441]. The new data in Supplemental Figure 3 are referred to in the Results section [Page 10, lines 204-207 and line 219] and the legend has been adjusted accordingly [legend included with figure].

Supplemental Figure 3. Small-interfering RNA (siRNA) can be used to knock down the expression of *unc-5* netrin receptor A (*UNC5A*), mitogen-activated protein kinase 1 (*MAPK1*), and nuclear factor kappa beta 1 (*NFKB1*) in equine mammosphere-derived epithelial cells (MDECs). **a)** Quantitative reverse transcription-polymerase chain reaction (qRT-PCR) analysis (**i**) and Western blot analysis (**ii**) of *UNC5A/UNC5a*, *MAPK1/ MAPK1* and *NFKB1/NF-κB* expression in equine MDECs transfected with a *UNC5A*, *MAPK1* and *NFKB1* siRNA cocktail or negative siRNA at the same concentration. **b), c), d)** qRT-PCR analysis (**i**) and Western blot analysis (**ii**) of either *UNC5A/UNC5a* (**b**), *MAPK1/ MAPK1* (**c**) or *NFKB1/NF-κB* (**d**) expression in equine MDECs transfected with either a *UNC5A*, *MAPK1*, or *NFKB1* siRNA or negative siRNA. Error bars show standard deviations. Different letters above bars indicate statistically significant differences. P < 0.05.

14. Assays that measure apoptosis rather than simply cell viability should be performed, and phase contrast pictures of these cells should be shown to the reader.
 >> *In line with this reviewer's comment and reviewer #2's comment 2, we carried out assays measuring apoptosis by determining caspase activity using flow cytometry. Moreover, and as*

requested, phase contrast images of the cells were taken prior to the flow cytometry assays. The capturing of these images has been added to the Materials and Methods section [Page 17, lines 374-375]. These data are shown in a new Supplemental Figure 4 and are referred to in the Results section [Page 10, lines 215-216 and Page 11, lines 223-226]. Also, a legend for this figure has been created [legend included with figure].

15. As a control, it would be appropriate to manipulate the expression of NFKB1 in canine MDECs to see if there is an effect on apoptosis. Moreover, there should be some insight into the mechanism of NFKB1 action in equine MDECs. Does DMBA treatment regulate NFKB expression in MDECs?

>> *In response to this comment, we performed a new set of experiments evaluating NFKB1 expression in both canine and equine MDECs under different conditions. The results from these experiments are shown in a new Figure 6 and described in a new paragraph in the Results section, entitled 'NFKB1 expression levels do not appear to modulate resistance of canine MDECs to DMBA-induced cell death, and decreased NFKB1 expression in equine MDECs relies on the combination of increased miR-214 expression and DMBA treatment' [Page 11-12, lines*

229-253]. A legend for this new figure has been created [Page 33, lines 720-730], and the results are now also briefly discussed in the Discussion section [Page 15, lines 318-321].

>> To respond to the reviewer's specific request to manipulate the expression of NFKB1 in canine MDECs to see if there is an effect on apoptosis, we want to mention that we did consider this. However, before manipulating/overexpressing NFKB1 in canine MDECs, we decided to first check the inherent NFKB1 expression in canine MDECs, which we expected to be lower compared to the inherent NFKB1 expression in equine MDECs based on our findings that: (i) miR-214 expression is inherently 'low' compared to miR-214 expression in canine MDECs and (ii) increasing miR-214 levels in equine MDECs leads to downregulation of NFKB1 expression. In contrast to this prediction, we found NFKB1 expression to be higher in canine MDECs compared to equine MDECs (Figure 6a), suggesting that NFKB1 is not involved in the response of canine MDECs to DMBA, and consequently, that manipulating NFKB1 in canine MDECs would not lead to an altered phenotype in response to DMBA.

16. Is NF- κ B or any of the candidate NF- κ B-regulated proteins — cyclin F (CCNF), cyclin E2 (CCNE2), colony stimulating factor 1 (CSF1) or the BRCA1-associated RING domain 1 (BARD1) involved in the DMBA-induced apoptotic response? And circling back to miR-214, where is this miR encoded in the genome and what are the regulatory mechanisms controlling its expression?

>> Although an interesting point raised by the reviewer related to potential involvement of NF- κ B-regulated proteins in the DMBA-induced apoptotic response, we believe that this is beyond the

scope of this work, which was initiated to determine which miRNAs modulate apoptosis of equine MDECs in response to DMBA. Still, and as requested, we have included information describing where miR-214 is predicted to be encoded in the human genome and which transcription factors could control its expression [Page 14, lines 308-312].

17. Minor pt. The figure is labelled Scramble siRNA but the manuscript refers to negative siRNA-transfected. It would be less confusing to keep the nomenclature similar between figure and manuscript.

>> Thank you for pointing this out and our apologies for this inconsistency. The text on the x-axes of the different graphs in Figure 5 and Supplemental Figures 4 & 5 has been corrected for consistency to negative siRNA.

Comments from Reviewer #2:

In this paper, the authors investigated miR-214-3p's role in DMBA-induced apoptosis of equine MDECs and observed that low levels of miR-214 promote NF κ B1 expression and thereby apoptosis in response to DMBA in equine MDECs. Further, it describes unc-5 netrin receptor A (UNC5A) as a downstream target of miR-214-3p for the first time. However, the conclusions drawn from the study are exaggerated and there are certain points that need to be addressed.

1. The role of NF κ B in apoptosis should be clearly described.

>> As requested, we now more clearly describe the role of NF κ B in apoptosis in general in the Discussion section [Page 14, lines 295-306]. We also conducted additional experiments to gain some insight into the mechanism of NF κ B1 in equine MDECs, as requested by reviewer #1 (see comment 15).

2. Cell viability study is insufficient to prove the DMBA-induced apoptosis in equine MDECs. Flow cytometry study should be conducted to confirm it.

>> As requested, and in line with a similar comment from reviewer #1 (see comment 14), we carried out assays measuring apoptosis by determining caspase activity using flow cytometry. The flow cytometry procedure has been added to the Materials and Methods section [Page 17, lines 370-375]. The new data in Figures 2 & 5 are referred to in the Results section [Page 7, lines 130-135; Page 10, lines 215-216; and Page 11, lines 223-224] and the legends have been adjusted accordingly [Page 31, lines 676-679 and Pages 32 and 33, lines 708-717].

3. Cell viability is insufficient to establish the involvement of NF κ B.

>> This comment is in line with the previous comment stating that assessing cell viability alone is insufficient, and should now be addressed by performing the requested flow cytometry assays evaluating apoptosis. These results are now included in Figure 5d(ii).

4. Units like micromolar (μ M) and micrometre (μ m) should be corrected.

>> Thank you for pointing this out. We have carefully reviewed the manuscript and corrected all unit abbreviations.

5. Objective and significance of this study is not clear. Describe it properly. Further, the paper should be written in a clear and concise way.

>> In response to the helpful comments and suggestions from both reviewers, we have substantially modified the manuscript. We believe that the implemented changes have increased the clarity on the objective and significance of our study.

Reviewers' comments:

Reviewer #1 (Remarks to the Author):

The authors did a thorough job addressing my comments and I am almost entirely satisfied with their responses except for one clarification about a new experiment (Figure 6) that was performed in response to both reviewers' comments. In my opinion, the authors need to better address their findings because they are confusing to me (see below). Also, the Western blots need to have MW markers and the full blots should be included in the supplementary information.

The authors conclude the manuscript by saying "Collectively, our data lead us to propose that relatively low levels of miR-214 allow for the robust expression of NFKB1, and apoptosis in response to DMBA in equine MDECs (Figure 7)." I understand why the authors want to conclude with this statement. But the last data experiment (Figure 6) is very confusing and in my view the results in this Figure do not support the authors' conclusions. Therefore this needs to be addressed.

The authors observe no change in NFKB1 expression in equine MDECs in the absence of DMBA +/- the miR-214 mimic (Figure 6c(i)). This is confusing to me but it presumably suggests that NFKB1 transcript levels are very low in the absence of DMBA so there is no reduction with the introduction of the miR-214 mimic. Unfortunately, we can't see this in the data because the data are normalized to 1 relative to the negative mimic. But let's say this is the case, then when is NFKB1 upregulated in equine MDECs in order to protect them from apoptosis in response to DMBA? Certainly, if they don't have constitutively high levels of NFKB1 expression, which the authors established they don't in Figure 6c(i), then we would expect that DMBA treatment would upregulate NFKB1. But this is not what the authors observe in Figure 6c(ii). Instead, they see no change in NFKB1 treatment in response to DMBA. So the question is: when do the authors see the "robust expression of NFKB1" that is presumably allowing apoptosis in response to DMBA. Why isn't it observed in Figure 6c(ii)? The authors conclude Figure 6 description by saying "Under both conditions, NFKB1 expression did not change, indicating that increased levels of miR-214 and DMBA treatment are needed together to decrease NFKB1 expression (Figure 3c & 6b) and prevent apoptosis of equine MDECs." But in my view, the authors find that NFKB1 expression does not appear to change in response to DMBA (Figure 6c(ii)), but it also never appears to be robust because in the absence of DMBA there isn't enough transcript to be downregulated by miR-214 (Figure 6c(i)). So how does NFKB1 protect the cells from apoptosis when the authors do not detect either upregulated in response to DMBA or robust constitutive expression of this gene?

We thank Reviewer #1 for thoroughly reading our revised manuscript. In addressing their queries, we believe that our manuscript has further improved in clarity. It is our hope that the revised manuscript will now be acceptable for publication in Communications Biology.

Comments from Reviewer #1:

The authors did a thorough job addressing my comments and I am almost entirely satisfied with their responses except for one clarification about a new experiment (Figure 6) that was performed in response to both reviewers' comments. In my opinion, the authors need to better address their findings because they are confusing to me (see below).

>> We are pleased that the reviewer is generally satisfied with our responses, and we greatly appreciate the time and effort they have invested to help us improve our manuscript. Please see our comments below regarding the experiments described in Figure 6 (comment #2).

1. The Western blots need to have MW markers and the full blots should be included in the supplementary information.

>> As requested, representative images of the full Western blots for each protein examined are now included in Supplemental Figure 3. Molecular weight markers are now shown. An updated description of this figure can be found on pages 10-11, lines 220-221, and a new legend is included as part of the figure.

Supplemental Figure 3. Small-interfering RNA (siRNA) were used to knock down the expression of unc-5 netrin receptor A (*UNC5A*), mitogen-activated protein kinase 1 (*MAPK1*), and nuclear factor kappa beta 1 (*NFKB1*) in equine mammosphere-derived epithelial cells (MDECs). a) Quantitative reverse transcription-polymerase chain reaction (qRT-PCR) analysis (i) and Western blot analysis (ii) of *UNC5A*/*UNC5A*, *MAPK1*/*MAPK1* and *NFKB1*/*NF-κB* expression in equine MDECs transfected with a *UNC5A*, *MAPK1* and *NFKB1* siRNA cocktail or negative siRNA at the same concentration. b), c), d) qRT-PCR analysis (i) and Western blot analysis (ii) of either *UNC5A*/*UNC5A* (b), *MAPK1*/*MAPK1* (c) or *NFKB1*/*NF-κB* (d) expression in equine MDECs transfected with either *UNC5A*, *MAPK1*, *NFKB1*, or negative siRNA. e) Representative images of Western blots for *UNC5A*, *MAPK1*, *NFKB1*, and reference protein β -actin. Error bars show standard deviations. Different letters above bars indicate statistically significant differences. $P < 0.05$.

2. The authors conclude the manuscript by saying "Collectively, our data lead us to propose that relatively low levels of miR-214 allow for the robust expression of NFKB1, and apoptosis in response to DMBA in equine MDECs (Figure 7)." I understand why the authors want to conclude with this statement. But the last data experiment (Figure 6) is very confusing and in my view the results in this Figure do not support the authors' conclusions. Therefore, this needs to be addressed.

>> We appreciate the opportunity to better present and explain the data in Figure 6. Please see our point-by-point responses below.

2.a. The authors observe no change in NFKB1 expression in equine MDECs in the absence of DMBA +/- the miR-214 mimic (Figure 6c(i)).

>> *That is correct.*

2.b. This is confusing to me but it presumably suggests that NFKB1 transcript levels are very low in the absence of DMBA so there is no reduction with the introduction of the miR-214 mimic. Unfortunately, we can't see this in the data because the data are normalized to 1 relative to the negative mimic.

>> *As the Reviewer suggests, it could be presumed that NFKB1 transcript levels are very low in the absence of DMBA, so there is no reduction with the introduction of the miR-214 mimic. However, we do not believe this is the case as the raw ct values for NFKB1 in equine MDECs determined by qRT-PCR, and although lower than the values seen in canine MDECs, are well within the acceptable range of a SYBR green-based assay. To illustrate that NFKB1 is constitutively and robustly expressed in both cell types, we have included a graph showing the raw ct values for NFKB1 in both equine and canine MDECs. This can be found in Figure 6a)(ii) and a description of this graph is now included on pages 11-12, lines 242-244, and in the legend for Figure 6 on page 35, lines 779-780.*

2.c. But let's say this is the case, then when is NFKB1 upregulated in equine MDECs in order to protect them from apoptosis in response to DMBA?

>> *Our conclusion is that "Collectively, our data lead us to propose that relatively low levels of miR-214 allow for the robust expression of NFKB1, and apoptosis in response to DMBA in equine MDECs (Figure 7)." We do not propose that the upregulation of NFKB1 prevents apoptosis of equine MDECs in response to DMBA, rather, we propose it **allows** apoptosis as a strategy to avoid potential cancer-causing mutations that might accumulate when damaged DNA is repaired.*

2.d. Certainly, if they don't have constitutively high levels of NFKB1 expression, which the authors established they don't in Figure 6c(i), then we would expect that DMBA treatment would upregulate NFKB1. But this is not what the authors observe in Figure 6c (ii). Instead, they see no change in NFKB1 treatment in response to DMBA. So, the question is: when do the authors see the "robust expression of NFKB1" that is presumably allowing apoptosis in response to DMBA. Why isn't it observed in Figure 6c(ii)? The authors conclude Figure 6 description by saying "Under both conditions, NFKB1 expression did not change, indicating that increased levels of miR-214 and DMBA treatment are needed together to decrease NFKB1 expression (Figure 3c & 6b) and prevent apoptosis of equine MDECs."

>> The raw *ct* values presented in new Figure 6a)(ii) indicates that *NFKB1* is constitutively and robustly expressed in equine MDECs (see above).

>> We have repeatedly observed that increased miR-214 and DMBA treatment are needed to decrease *NFKB1* expression. We do not know why both conditions are required, and we believe that it is beyond the scope of this study to explore that mechanism. Still, we now propose a potential explanation in the Discussion section, pages 15-16, lines 322-354, and we also made a graphical abstract depicting this potential mechanism to clarify the text visually. This can be found as a new Supplemental Figure 5. After we created this Supplemental Figure, we updated Figure 7 for consistency. An updated description of this figure is included in the Conclusion on pages 16-17, lines 357-362. The legend for Figure 7 on pages 35-36, lines 785-796, has been rewritten to more accurately reflect what is now shown.

Supplemental Figure 5. A potential mechanism by which microRNA-214 (miR-214) and 7, 12-Dimethylbenz(a)anthracene (DMBA) decrease nuclear factor kappa beta 1 (*NFKB1*) expression in equine mammosphere derived epithelial cells (MDECs). **a)** At baseline, miR-214 expression is low, and a miRNA sponge blocks its function, allowing for robust expression of target *NFKB1*. **b)** Addition of a miR-214 mimic creates a positive feedback loop with increased expression of the miR-214 sponge, which binds to and blocks the activity of the supplementary miR-214. Expression of *NFKB1* continues to be robust as miR-214 activity is blocked. **c)** DMBA treatment alone does not affect *NFKB1* expression. **d)** When addition of a miR-214 mimic is followed by DMBA treatment, the positive feedback loop between miR-214 and the miR-214 sponge is disrupted by DMBA. miRNA-214 activity is no longer blocked and the excess of free miR-214 can interact with the 3'UTR of *NFKB1*, resulting in decreased *NFKB1* expression.

2.e. But in my view, the authors find that *NFKB1* expression does not appear to change in response to DMBA (Figure 6c(ii)), but it also never appears to be robust because in the absence of DMBA there isn't enough transcript to be downregulated by miR-214 (Figure 6c(i)). So how does *NFKB1* protect the cells from apoptosis when the authors do not detect either upregulated in response to DMBA or robust constitutive expression of this gene?

>> Again, we do consider constitutive expression of *NFKB1* to be robust in equine MDECs, as now shown by the raw *ct* values in Figure 6 a)(ii). Based the results from experiments where both the miR-214 mimic and DMBA were used (Figure 2) combined with the results from experiments where siRNA was used to knock down expression of *NFKB1* (Figure 5d)(i)&(ii)), we propose that

this constitutive expression is adequate to allow equine MDECs to undergo apoptosis, as a mechanism to protect them from mutations that may arise from repairing DMBA-induced DNA damage (Figure 7).

>> In addition to the newly added information on the constitutive expression of NFKB1 in equine MDECs on pages 11-12, lines 242-244, we added the word “constitutive” to the abstract (page 2, line 34) to clarify that the inherent level of NFKB1 in equine MDECs seems to modulate the apoptotic response induced by DMBA.

a) DMBA

b) miRNA 214 ↑ + DMBA

c) NFKB1 ↓ + DMBA

Figure 7. Harman et al.

REVIEWERS' COMMENTS:

Reviewer #1 (Remarks to the Author):

The authors have addressed my concerns about figure 6 by proposing the highly speculative “miR-214 sponge” model that could explain why NFkB1 is not regulated by the miR-214 mimic in equine MDECs in the absence of DMBA. The model proposes that DMBA disrupts this as yet unidentified “sponge”, and this allows miR-214 accumulation that decreases NFkB expression, preventing equine MDECs from undergoing apoptosis and leaving them vulnerable to oncogenic mutations due to the DMBA treatment. There is a lot that is still unknown. I think, however, this model will help readers put the data in a potential biological context. At the same time, the speculative model also makes it very clear that further investigation is necessary to fully understand the mechanism of NFkB regulation and, more importantly, mammary cancer resistance mechanisms.